# A FRET-based respirasome assembly screen identifies spleen tyrosine kinase as a target to improve muscle mitochondrial respiration and exercise performance in mice

Ami Kobayashi [1,2], Kotaro Azuma[1], Toshihiko Takeiwa[1], Toshimori Kitami [3], Kuniko Horie [4], Kazuhiro Ikeda [4] & Satoshi Inoue [1,4] ✉

Aerobic muscle activities predominantly depend on fuel energy supply by mitochondrial respiration, thus, mitochondrial activity enhancement may become a therapeutic intervention for muscle disturbances. The assembly of mitochondrial respiratory complexes into higher-order "supercomplex" structures has been proposed to be an efficient biological process for energy synthesis, although there is controversy in its physiological relevance. We here established Förster resonance energy transfer (FRET) phenomenon-based live imaging of mitochondrial respiratory complexes I and IV interactions using murine myoblastic cells, whose signals represent in vivo supercomplex assembly of complexes I, III, and IV, or respirasomes. The live FRET signals were well correlated with supercomplex assembly observed by blue native polyacrylamide gel electrophoresis (BN-PAGE) and oxygen consumption rates. FRET-based live cell screen defined that the inhibition of spleen tyrosine kinase (SYK), a non-receptor protein tyrosine kinase that belongs to the SYK/ zeta-chain-associated protein kinase 70 (ZAP-70) family, leads to an increase in supercomplex assembly in murine myoblastic cells. In parallel, SYK inhibition enhanced mitochondrial respiration in the cells. Notably, SYK inhibitor administration enhances exercise performance in mice. Overall, this study proves the feasibility of FRET-based respirasome assembly assay, which recapitulates in vivo mitochondrial respiration activities.

In eukaryotic cells, mitochondria generate ATP through a process called oxidative phosphorylation (OXPHOS), which is carried out by five enzymatic complexes in the inner mitochondrial membrane (complex I–IV and ATP synthase) and two mobile electron carriers (coenzyme Q and cytochrome *c*). Complexes I–IV, called mitochondrial respiratory chain (MRC) complexes, carries protons across the inner mitochondrial membrane, which becomes a driving force for ATP generation by ATP synthase. These MRC complexes can form higher-order structures called MRC supercomplexes, consisting of complexes I, III, and IV in mammalian cells[1,2]. The physiological significance of the MRC supercomplexes are still in debate. Some reports suggest that the MRC supercomplex assembly enhances efficacy of

[1]Department of Systems Aging Science and Medicine, Tokyo Metropolitan Institute of Gerontology, 35-2 Sakae-cho, Itabashi-ku, Tokyo 173-0015, Japan. [2]Department of Geriatric Medicine, Graduate School of Medicine, The University of Tokyo, 7-3-1 Hongo, Bunkyo-ku, Tokyo 113-8655, Japan. [3]Laboratory for Metabolic Networks, RIKEN Center for Integrative Medical Sciences, 1-7-22 Suehiro-cho, Tsurumi-ku, Yokohama, Kanagawa 230-0045, Japan. [4]Division of Systems Medicine and Gene Therapy, Saitama Medical University, 1397-1 Yamane, Hidaka-shi, Saitama 350-1241, Japan. ✉e-mail: sinoue@tmig.or.jp

respiratory chain reaction by shortening the distance between respiratory complexes[3,4], by modulating the assembly and stability of complex I[5], or by changing the structural surface for quinone-binding sites[6]. In addition, substrate channeling and sequestration of quinone pool are proposed[1,7–10] as important changes induced by MRC supercomplexes. However, recent biochemical and structural data question these hypotheses[11–14]. MRC supercomplex assembly also minimizes excessive ROS production[8,15,16] although this point still remains controversial. Furthermore, the "plasticity model" of MRC has been proposed in which the MRC supercomplexes contribute to metabolic adaptation by rearranging themselves in response to changes in metabolic source of electrons[16–20]. All of these point to the importance of developing new tools for studying and manipulating MRC supercomplexes.

Among different supercomplexes, the respirasome is the major structure that consists of complex I and complex IV monomers with a complex III dimer (I/III$_2$/IV), accounting for over 50 percent of all supercomplex structures in mammalian cells[2]. In addition, a complex III dimer forms a supercomplex with complex I monomer (I/III$_2$) or with complex IV monomer (III$_2$/IV). Complex II and ATP synthase have not been shown to physically bind to other complexes in mammalian cells[21].

We and others previously identified a mitochondrial protein, cytochrome $c$ oxidase subunit 7a-related polypeptide (COX7RP, also known as COX7A2L/SCAF1), as an MRC supercomplex assembly factor[7,22–27], which was originally discovered as an estrogen-inducible protein[28]. The precise role of COX7RP, that is, whether COX7RP is involved in the formation of supercomplexes I/III$_2$/IV and III$_2$/IV or stabilizes supercomplex III$_2$/IV without affecting supercomplex I/III$_2$/IV, is still highly debated[7,20,22,24–27,29–33].

In addition to COX7RP, several proteins have been identified as respiratory supercomplex assembly factors[34]. However, the molecular pathways regulating MRC supercomplex formation are largely unknown. While blue native polyacrylamide gel electrophoresis (BN-PAGE) or cryo-electron microscopy are usually used for analyzing the status of MRC supercomplexes, it will be further beneficial to develop an alternative rapid and simple quantitative assay for the evaluation of MRC supercomplex status, which can be applied to a medium-throughput or high-throughput screening. Moreover, the identification of factors involved in promoting supercomplex assembly holds therapeutic promise for diseases such as sarcopenia[35]. For example, aging decreases MRC supercomplex formation in rat hearts[36] and overexpression of MRC assembly factor COX7RP enhances running capacity in transgenic mice[22]. These studies point to a need for a higher-throughput approach in assaying respiratory supercomplex formation for uncovering molecular pathways and therapeutics of MRC supercomplex.

Here, we developed a screening assay for MRC supercomplex formation by applying Förster resonance energy transfer (FRET) phenomenon with improved throughput compared to BN-PAGE method. FRET involves energy transfer between fluorophores[37] and is sensitive to small changes in distance[38]. FRET has been applied to monitor protein interactions in complexes and re-location of proteins between cellular micro-compartments[39]. We generated intermolecular FRET probes between complex I and IV and generated a scoring system to screen for MRC supercomplex assembly. Using this FRET-based method, we screened over 1200 bioactive small molecules and identified SYK inhibitors as enhancers of MRC supercomplex formation, which was validated using BN-PAGE, and genetically confirmed with siRNA. In vivo, SYK inhibitors enhanced MRC respiratory capacity and enhanced exercise performance in mice. Our FRET-based screening assay and small molecule enhancers of MRC supercomplex formation hold promise in uncovering additional regulatory pathways and in identifying novel therapeutics surrounding MRC supercomplex biology.

## Results

### FRET biosensors of mitochondrial respiratory complexes enable visualization of supercomplex formation in myoblastic cells

To develop a FRET-based assay of MRC supercomplex formation, we focused on the interaction between respiratory complexes I and IV. Previous literature has defined that complexes I/III$_2$/IV, I/III$_2$, and III$_2$/IV are predominant MRC supercomplexes in mammalian cells. While a possibility remains for the existence of other types of supercomplex such as I$_2$/III$_2$/IV$_2$ in humans and I/II/III$_2$/IV$_2$ in ciliates[40,41], we assume that live monitoring of complexes I and IV interaction mostly reflect the status of physical interactions of complexes I/III$_2$/IV in living mammalian cells.

Based on a previous structural study of MRC supercomplex at near-atomic resolution[42], we selected NDUFB8 from complex I and COX8A from complex IV for FRET-based assay as these subunits become physically close to each other upon supercomplex formation. We generated AcGFP-tagged NDUFB8 (AcGFP-NDUFB8) and DsRed-Monomer-tagged COX8A (DsRed-COX8A), which were stably co-expressed in murine C2C12 myoblastic cells. Upon supercomplex formation, the excitation of AcGFP-tagged complex I should result in an emission signal from DsRed-Monomer-tagged complex IV (Fig. 1a). We detected emission signal from DsRed-Monomer upon AcGFP excitation in fixed C2C12 myoblastic cells stably co-expressing AcGFP-NDUFB8 and DsRed-COX8A (Fig. 1b). To ensure that the FRET signals were specific to supercomplex formation, we also generated FRET-based assay between complex III and ATP synthase (complex V), which do not form supercomplex. The emission signals from DsRed-Monomer were not detected in fixed C2C12 myoblastic cells stably co-expressing AcGFP-tagged UQCR11 (a subunit of complex III) and DsRed-Monomer-tagged ATP5F1c (a subunit of ATP synthase) upon AcGFP excitation (Fig. 1b). We also confirmed exclusive localization of fluorophores in mitochondria (Supplementary Fig. 1a–d). Considering that ATP synthase is not a component of the mitochondrial respiratory chain supercomplex, the emission signal of DsRed-Monomer may reflect the FRET phenomenon. We also detected the emission signal of DsRed-Monomer in live C2C12 myoblastic cells stably co-expressing AcGFP-NDUFB8 and DsRed-COX8A (Fig. 1c), but not in live C2C12 myoblastic cells stably co-expressing AcGFP-tagged UQCR11 and DsRed-Monomer-tagged ATP5F1c (Fig. 1c). To verify that the emission signals of DsRed-Monomer were specific to the FRET phenomenon, we photobleached DsRed-Monomer in several areas (Fig. 1d). The photobleaching of DsRed-Monomer enhanced AcGFP fluorescence signal (Fig. 1e), which confirms that AcGFP and DsRed-Monomer functioned as a donor and an acceptor in FRET-based assay.

### Corrected FRET signal normalized to donor signal reflects MRC supercomplex assembly

To functionally validate whether the FRET signals between complexes I and IV detect qualitative changes in MRC supercomplex formation, we silenced the expression of supercomplex assembly factor COX7RP using specific small interfering RNAs (siRNAs) (siCox7rp #1 and #2) (Fig. 2a, b). We first validated the downregulation of MRC supercomplex formation in cells transfected with siCox7rp using BN-PAGE assay. Consistent with previous reports[22], the suppression of COX7RP expression resulted in decreased supercomplex formation in C2C12 myoblastic cells (Fig. 2c). We next silenced COX7RP in C2C12 myoblastic cells stably co-expressing AcGFP-NDUFB8 and DsRed-COX8A with siCox7rp and performed qualitative analysis using fluorescence microscopy under live conditions. We observed lower FRET efficiency in cells treated with siCox7rp compared to cells treated with control siRNAs (siControl #1 and #2) (Fig. 2d). These results confirm that our FRET-based assay detects MRC supercomplex formation. To convert our FRET-based MRC supercomplex formation assay into a medium-throughput

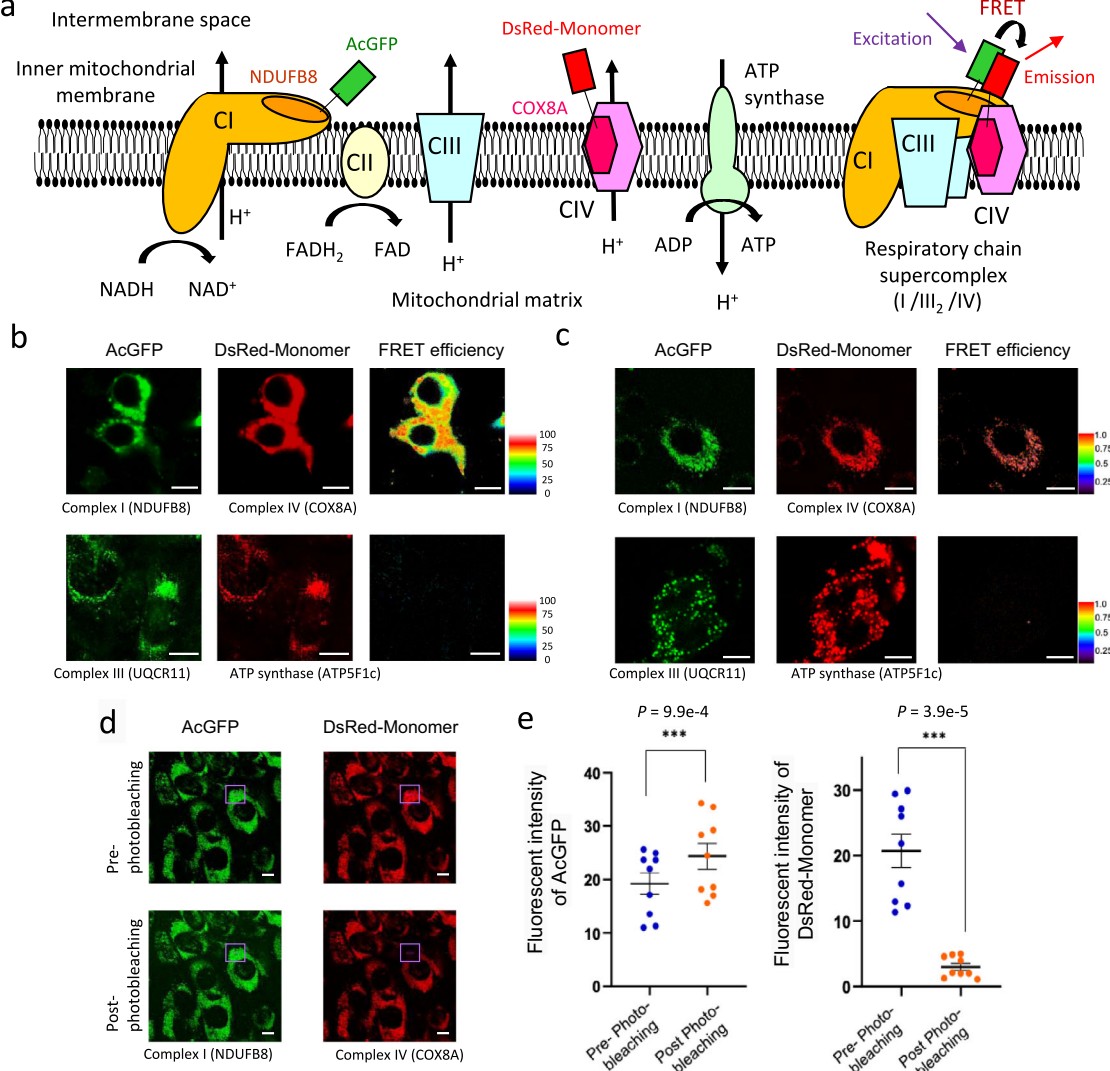

**Fig. 1 | Intermolecular FRET biosensors of mitochondrial respiratory chain complex detect supercomplex formation in fixed and live cells. a** Schema of five molecular complexes involved in oxidative phosphorylation; complex I–IV and ATP synthase. A subset of mitochondrial respiratory chain complexes forms a higher-order structure called supercomplex. The most abundant supercomplex consisting of complex I (CI) monomer, complex III (CIII) dimer, and complex IV (CIV) monomer (I/III₂/IV) is shown. Förster resonance energy transfer (FRET) biosensors of AcGFP tethered to NDUFB8 (a subunit of CI) and DsRed-Monomer tethered to COX8A (a subunit of CIV) are shown. **b, c** Fluorescence microscopy images of C2C12 myoblastic cells stably co-expressing NDUFB8-AcGFP and COX8A-DsRed-Monomer (upper panels), and UQCR11-AcGFP and ATP5F1c-DsRed-Monomer (lower panels),

in fixed (**b**) and live (**c**) cells. FRET efficiency is shown in pseudo-color image. These experiments were repeated twice and the results of one experiment are shown. Scale bars; 10 μm. **d** Fluorescence images of acceptor-photobleaching. In C2C12 myoblastic cells stably expressing NDUFB8-AcGFP and COX8A-DsRed-Monomer, DsRed-Monomer was partially photobleached by illumination at 558 nm for 3 min. The photobleached area (purple square) is indicated. Scale bars; 10 μm.
**e** Fluorescence images of DsRed-Monomer and AcGFP channels before and after photobleaching. Fluorescence intensities were measured in selected regions (*n* = 9) and compared before and after bleaching. Data are presented as means ± SE.
***P < 0.001; paired two-sided Student's *t* test. Source data are provided as a Source Data file.

---

screening assay, we quantified FRET signals using an imaging cytometer. C2C12 myoblastic cells expressing both AcGFP-NDUFB8 and DsRed-COX8A were treated with siCox7rp, and the fluorescence signals of AcGFP and DsRed-Monomer and the raw FRET signal (signal of DsRed-Monomer when AcGFP was excited) were measured under live conditions. The fluorescence signal of AcGFP (Fig. 2e) or DsRed-Monomer (Fig. 2f) did not change when the cells were treated with siCox7rp. However, when FRET signals were normalized to the donor signal as corrected FRET per donor (cFRET/donor) using previously described normalization method[43–47], siCox7rp treatment significantly decreased the cFRET/donor value (Fig. 2g). These results indicated that the cFRET/donor value calculated from the quantified FRET signal reflected the alteration of MRC supercomplex formation.

**FRET-based medium-throughput screen identifies chemical compounds promoting supercomplex assembly**

β-Lapachone is a naturally occurring benzochromenone including heterotricyclic ring system obtained from the bark of the lapacho tree (*Tabebuia avellanedae*) (Supplementary Fig. 2a), and known to alleviate the decrease in the number of mitochondria and abnormality in mitochondrial structure in skeletal muscle of aged mice, and to enhance energy metabolism, muscle strength, and exercise capacity[48]. In addition, it was reported that β-lapachone treatment increased the expression of some mitochondria-related genes including genes encoding complex IV subunits and enhanced mitochondrial function in adipocytes[49]. We thus hypothesized that β-lapachone is involved in MRC supercomplex formation and examined this hypothesis using our FRET-based quantification method of MRC supercomplex. C2C12

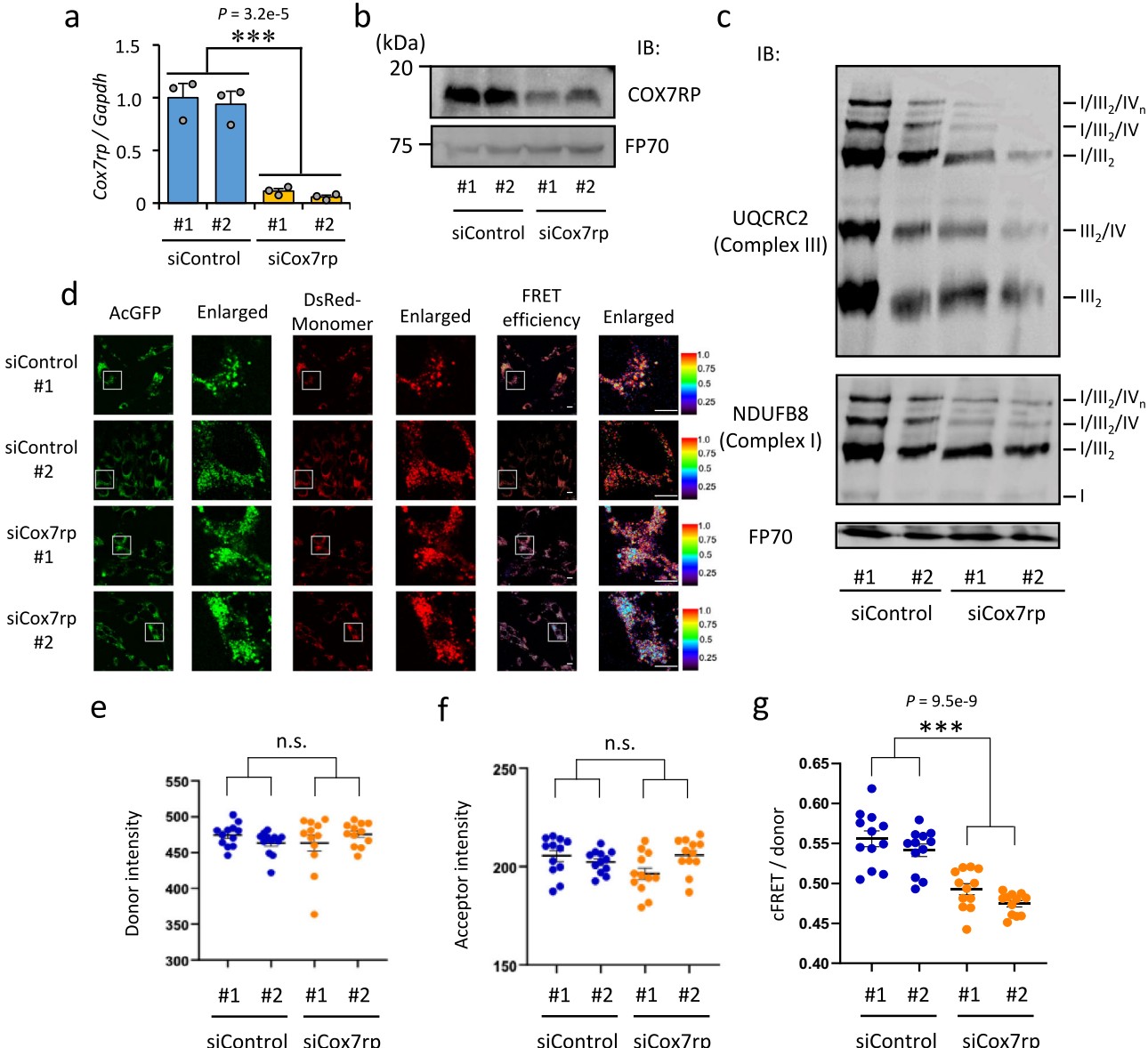

**Fig. 2 | Intermolecular FRET biosensors of mitochondrial respiratory chain complex detect reduced supercomplex formation by suppressing Cox7rp expression. a** Knockdown of *Cox7rp* expression with siCox7rp in C2C12 myoblastic cells was performed by reverse transfection method. Two days after transfection, total RNA was extracted and knockdown efficiency was evaluated using qRT-PCR. Two different siRNAs (10 nM) targeting *Cox7rp* (siCox7rp #1 and #2) and two different siRNAs (10 nM) not targeting human transcripts (siControl #1 and #2) were used. Data are presented as means ± SE (*n* = 3 biologically independent samples). ***P < 0.001; two-way ANOVA. **b** Knockdown efficiency of COX7RP in C2C12 myoblastic cells evaluated by western blot analysis. Knockdown of *Cox7rp* expression with siCox7rp in C2C12 myoblastic cells was performed by reverse transfection method. Two days after transfection, the cells were lysed and subjected to western blot analysis with the COX7RP antibody. FP70 protein was blotted as an internal control. This experiment was repeated twice and the results of one experiment are shown. IB, immunoblot. **c** Mitochondrial proteins of C2C12 myoblastic cells treated with siCox7rp #1 or #2, or

siControl #1 or #2 were solubilized and subjected to blue native polyacrylamide gel electrophoresis (BN-PAGE). Positions corresponding to mitochondrial supercomplex I/III$_2$/IV$_n$, I/III$_2$/IV, I/III$_2$, III$_2$/IV, complex I, and dimerized complex III (III$_2$) are indicated. BN-PAGE was performed with antibodies for NDUFB8 of complex I and UQCRC2 of complex III. FP70 protein was blotted as an internal control. This experiment was repeated twice and the results of one experiment are shown. **d** Fluorescence microscopy images of C2C12 myoblastic cells stably co-expressing NDUFB8-AcGFP and COX8A-DsRed-Monomer, treated with siCox7rp or siControl in live cells. FRET efficiency is shown by pseudo-color images. Indicated area (white square) are shown in the enlarged images. Scale bars, 10 μm. **e–g** Quantification of fluorescence intensities of C2C12 myoblastic cells stably co-expressing NDUFB8-AcGFP and COX8A-DsRed-Monomer, treated with siCox7rp or siControl by imaging cytometer. Donor intensities (**e**), acceptor intensities (**f**), and corrected FRET (cFRET)/donor ratios (**g**) are shown. Data from 12 wells are presented as means ± SE. n.s. not significant. ***P < 0.001, two-way ANOVA. Source data are provided as a Source Data file.

myoblastic cells expressing both AcGFP-NDUFB8 and DsRed-COX8A were treated with β-lapachone and FRET signals were measured using an imaging cytometer. The value of cFRET/donor significantly increased by treatment of β-lapachone at a concentration of 1 μM, while vehicle (DMSO) treatment had no significant effect (Supplementary Fig. 2b, c). In addition, cFRET/donor value increased in a concentration-dependent manner at a concentration of 0.025-5 μM,

suggesting β-lapachone as a candidate compound to promote MRC supercomplex formation (Supplementary Fig. 2d).

Next, to comprehensively search for compounds that promote MRC supercomplex formation, we applied our quantitative FRET-based method to a medium-throughput chemical screen for MRC supercomplex assembly (Fig. 3a). C2C12 myoblastic cells stably co-expressing AcGFP-NDUFB8 and DsRed-COX8A were treated with a

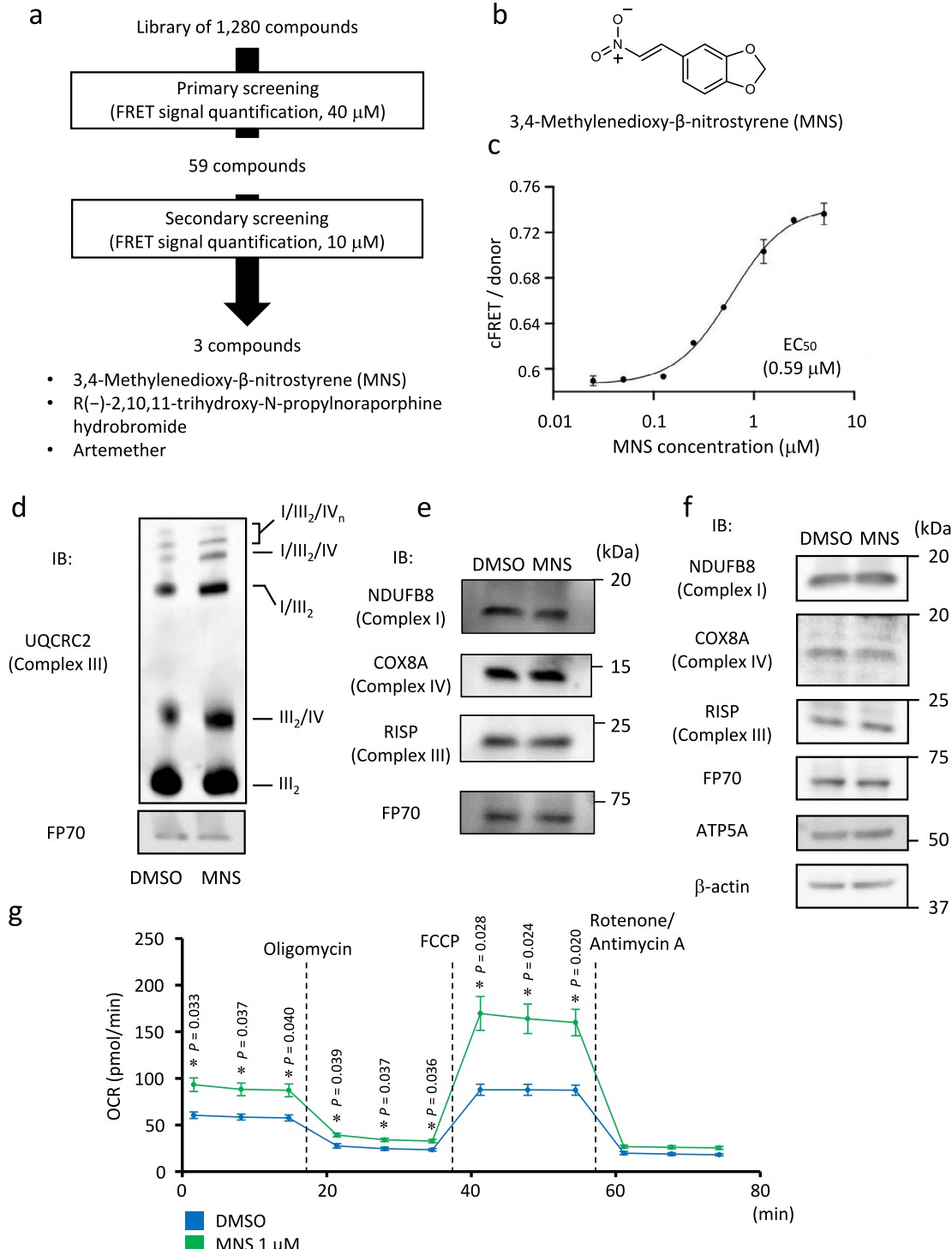

**Fig. 3 | 3,4-methylenedioxy-β-nitrostyrene (MNS) was identified as a compound inducing mitochondrial respiratory chain supercomplex formation using a chemical library screen with FRET imaging. a** Scheme of the medium-throughput screen procedures using imaging cytometer. Compounds that induced high cFRET/donor value in C2C12 cells stably co-expressing NDUFB8-AcGFP and COX8A-DsRed-Monomer were selected for further analysis. **b** Chemical structures of MNS. **c** cFRET/donor ratio of C2C12 cells stably co-expressing NDUFB8-AcGFP and COX8A-DsRed-Monomer treated by different concentrations of MNS for 24 h. Data are presented as means ± SE of three wells for each treatment. $EC_{50}$, half maximal effective concentration. **d** BN-PAGE of mitochondrial proteins from C2C12 cells treated with MNS (1 μM) or DMSO for 24 h. Positions corresponding to indicated mitochondrial supercomplexes and dimerized complex III ($III_2$) are indicated. Immunoblot (IB) was probed with anti-UQCRC2. FP70 protein was analyzed as an internal control. **e** SDS-PAGE of mitochondrial fraction from C2C12 cells with treatment as panel **d**. IB was probed with antibodies against distinct respiratory complexes. **f** SDS-PAGE of whole cell lysates from C2C12 cells with treatment as panel **d**. IB was probed with indicated antibodies. β-Actin was probed as an internal control. **g** Oxygen consumption rate (OCR) measurement of C2C12 cells with treatment as panel **d** using Seahorse XFp Cell Mito Stress Test. Data are presented as means ± SE from three biologically independent experiments. Basal respiration, ATP synthesis component of respiration, and maximal respiration were calculated as described in "Methods". FCCP, carbonyl cyanide 4-(trifluoromethoxy) phenylhydrazone. *$P < 0.05$; unpaired two-sided Student's $t$ test. Source data are provided as a Source Data file.

library of 1280 pharmacologically active chemicals (LOPAC 1280), which is widely used to screen small chemicals that modulate various biological events[50–52]. In the primary screen, the FRET signals in cells treated with these chemicals (40 μM each), β-lapachone (40 μM), and vehicle (DMSO) were evaluated using an imaging cytometer. The concentration of compounds was determined based on our previous experience using a LOPAC 1280 library[50]. FRET signals were measured in two replicate experiments and the values were calculated for each compound using the following formula: (mean cFRET/donor of compound − mean cFRET/donor of DMSO)/(mean cFRET/donor of β-lapachone − mean cFRET/donor of DMSO).

We focused on 222 compounds with values higher than 0.6, and finally selected 59 compounds by excluding those with prominent cell death, abnormal cell morphology, or a decrease in cell number, and those with cFRET/donor values that differed by more than 0.15 between two measurements (Supplementary Data 1).

In the secondary screen, we examined cFRET/donor value in cells treated with 59 compounds at a concentration of 10 μM using our quantitative FRET-based assay using imaging cytometer. FRET signals were measured using an imaging cytometer in two replicate experiments and calculated the value for each compound using the following formula: mean cFRET/donor of compound − mean cFRET/donor of DMSO. Among these compounds, 3,4-methylenedioxy-β-nitrostyrene (MNS), R( − )−2,10,11-trihydroxy-N-propylnoraporphine hydrobromide, and artemether were top 3 compounds exhibiting the highest values with 1.65, 1.07, and 0.78, respectively (Supplementary Data 2). Thus, we focused on MNS[53], a cell-permeable β-nitrostyrene derivative that functions as a SYK inhibitor, and performed further analysis in the present study (Fig. 3b).

Based on dose-dependent alterations of cFRET/donor values, we estimated the $EC_{50}$ value of MNS as 0.59 μM (Fig. 3c). We evaluated MRC supercomplex formation based on BN-PAGE for three times and identified MNS as a chemical that promotes supercomplex formation (Fig. 3d and Supplementary Fig. 3a, b). We found that protein levels of NDUFB8 (complex I), RISP (complex III), and COX8A (complex IV) in the mitochondrial fraction and whole cell lysates were not markedly changed in response to MNS (Fig. 3e, f), suggesting that increased abundance of mitochondria, or mitochondrial biogenesis, may not be the primary reason for increased MRC supercomplex formation. In addition, MNS elevated both basal and maximal oxygen consumption rates (OCRs) in C2C12 myoblastic cells, while it had no significant effect on extracellular acidification rates (ECARs) in those cells. (Fig. 3g and Supplementary Fig. 4a), suggesting that MRC supercomplex formation by MNS also leads to functional changes to respiratory chain function.

## Inhibition of SYK increases respiratory chain supercomplex assembly

In order to ensure that enhanced MRC supercomplex formation by MNS was due to on-target effect, we tested other SYK inhibitors for MRC supercomplex formation. We treated C2C12 myoblastic cells stably co-expressing AcGFP-tagged NDUFB8 and DsRed-Monomer-tagged COX8A with SYK inhibitors BAY61-3606 and GSK143 (Fig. 4a). We observed dose-dependent increases in cFRET/donor values for both of these SYK inhibitors, and $EC_{50}$ values of BAY61-3606 and GSK143 were estimated as 1.12 μM and 1.64 μM, respectively (Fig. 4b, c). We performed BN-PAGE for three times and showed that BAY61-3606 and GSK143 enhance MRC supercomplex formation (Fig. 4d and Supplementary Fig. 3a, c). These additional SYK inhibitors also did not obviously increase the protein abundance of respiratory chain subunits in the mitochondrial fraction and whole cell lysates (Fig. 4e, f). Moreover, similar to MNS, BAY61-3606 increased both basal and maximal oxygen consumption rates (Fig. 4g). For GSK143, we detected increases only in maximal oxygen consumption rates (Fig. 4h). In addition, GSK143 did not significantly affect the ATP synthesis component of respiration (Fig. 4h). Treatment of BAY61-

3606 and GSK143 did not affect ECARs significantly (Supplementary Fig. 4b, c). These results suggest that variety of SYK inhibitors can enhance MRC supercomplex formation in C2C12 myoblastic cells.

To genetically validate the on-target effect of SYK protein on MRC supercomplex formation, we downregulated SYK expression using siRNA (Fig. 5a, b). Transfection of siRNA targeting SYK in C2C12 myoblastic cells stably co-expressing AcGFP-tagged NDUFB8 and DsRed-Monomer-tagged COX8A showed enhanced cFRET/donor values (Fig. 5c). We performed BN-PAGE for three times and showed that siSyk #1 and #2 treatment enhanced MRC supercomplex formation in C2C12 myoblastic cells (Fig. 5d and Supplementary Fig. 3d, e). The protein concentrations of respiratory chain subunits in whole cell lysates were not markedly changed by silencing SYK expression (Fig. 5b). Furthermore, siSyk #1 and #2 treatment resulted in enhanced maximal oxygen consumption rates, while did not significantly alter the ATP synthesis component of respiration and ECARs was detected (Fig. 5e, f and Supplementary Fig. 4d, e). These results confirm that SYK inhibition promotes MRC supercomplex formation.

## Mice treated with SYK inhibitors display enhanced exercise performance

To assess the physiological relevance of SYK inhibitors in vivo, we intraperitoneally administered MNS to 2-month-old mice for 5 weeks. There was no significant change in body weight between vehicle (DMSO)- and MNS-treated mice (Fig. 6a). After 3 weeks of treatment, MNS-treated mice displayed longer hanging time compared to vehicle-treated mice in a wire hang test (Fig. 6b). In a treadmill test, longer running distances and times were observed in the MNS-treated mice compared to vehicle-treated mice (Fig. 6c, d and Supplementary Movie 1). The measurement of oxygen consumption during exercise 4 weeks after MNS administration revealed higher oxygen consumption rates in MNS-treated mice compared to control mice (Fig. 6e). Dissected muscle weights of hind limbs 5 weeks after MNS administration were not significantly altered (Supplementary Fig. 5a–c). In the mitochondrial fraction of quadriceps femoris muscles from MNS-treated mice, respiratory supercomplex assembly was substantially enhanced as analyzed by BN-PAGE (Fig. 6f). Ultrastructural analysis revealed that mitochondrial size in soleus muscles was not markedly different between the muscles from MNS-treated mice and DMSO-treated mice (Supplementary Fig. 5d, e). The concentrations of respiratory chain subunits were also unchanged by MNS treatment as analyzed by SDS-PAGE (Fig. 6g). These results suggest that MNS enhance MRC supercomplex formation in vivo.

We also evaluated whether two additional SYK inhibitors, BAY61-3606 and GSK143, affected the in vivo function of skeletal muscles by administrating these inhibitors to 2-month-old mice. No significant differences were observed in the body weights of mice treated with vehicle (DMSO) or these inhibitors for 6 weeks (Supplementary Fig. 6a). In a wire hang test performed 3 weeks after drug administration, the hanging time of SYK inhibitor-treated mice was longer than that of the control mice (Supplementary Fig. 6b). There was no significant difference in hanging time between mice treated with DMSO alone and physiological saline (Supplementary Fig. 6c). In a treadmill test performed 4 weeks after the drug administration, longer running distances and times were observed in the SYK inhibitor-treated mice than in the control mice (Supplementary Fig. 6d, e and Supplementary Movie 2). The weight of the dissected muscles (Supplementary Fig. 6f–h) was not significantly changed after the 6-week administration of either compound. In the mitochondrial fraction of quadriceps femoris muscles from SYK inhibitor-treated mice, respiratory supercomplex assembly was substantially enhanced as analyzed by BN-PAGE (Supplementary Fig. 6i), while the protein concentrations of respiratory chain subunits were not markedly changed (Supplementary Fig. 6j). Ultrastructural analysis showed that mitochondrial size in the soleus muscles was not obviously different between vehicle- and SYK

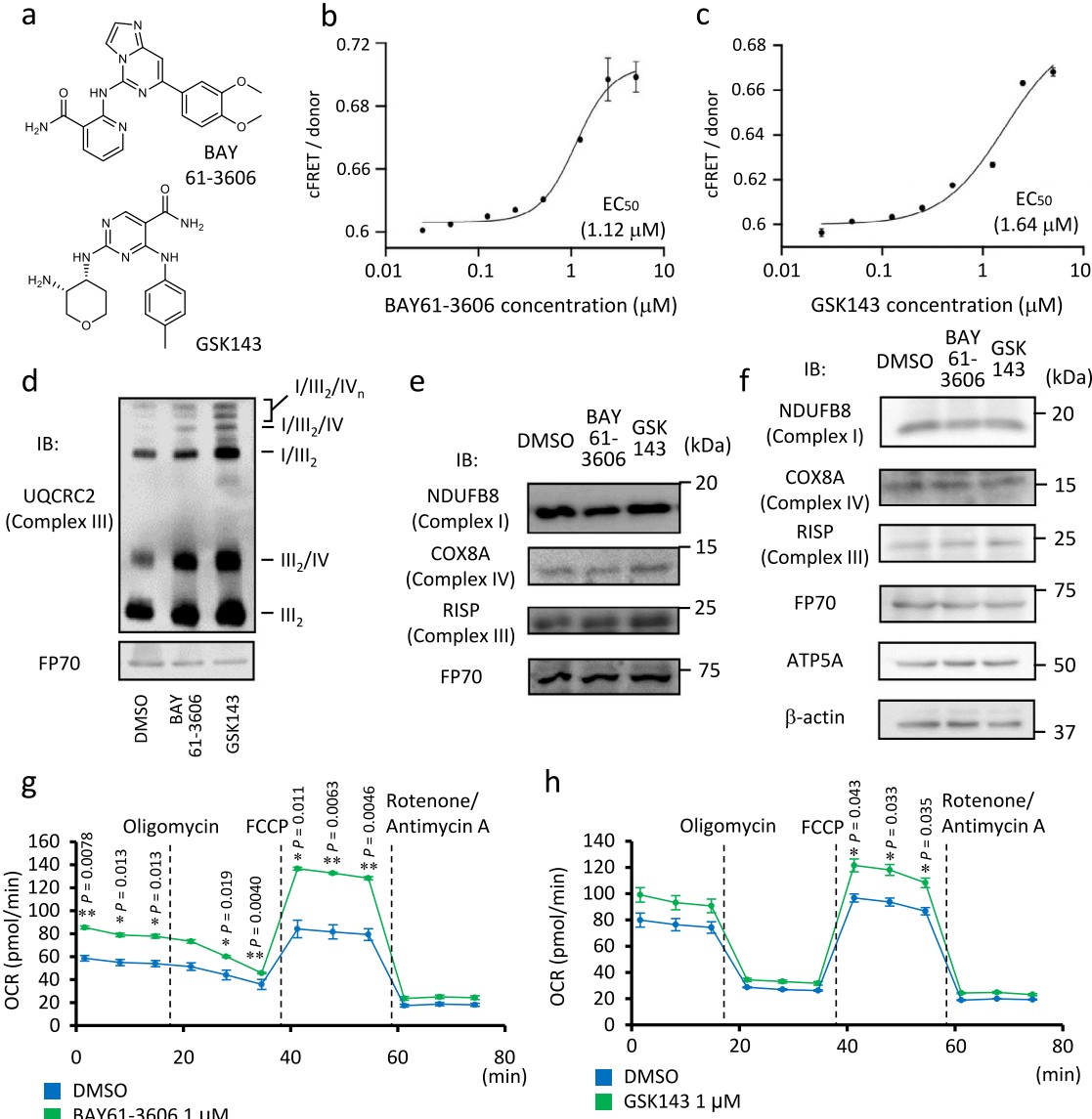

**Fig. 4 | SYK inhibitors promote mitochondrial respiratory chain supercomplex assembly and stimulates oxygen consumption in C2C12 myoblastic cells.**
**a** Chemical structures of BAY61-3606 and GSK143. **b**, **c** cFRET/donor ratio of C2C12 cells stably co-expressing NDUFB8-AcGFP and COX8A-DsRed-Monomer treated by different concentrations of BAY61-3606 (**b**) and GSK143 (**c**). Data are presented as means ± SE of three wells for each treatment. **d** BN-PAGE of mitochondrial proteins from C2C12 cells treated with indicated SYK inhibitors (1 μM each) or DMSO for 24 h. Positions corresponding to indicated mitochondrial supercomplexes and dimerized complex III (III$_2$) are indicated. Immunoblot (IB) was probed with anti-UQCRC2. FP70 was analyzed as an internal control. **e** IB for a mitochondrial fraction of C2C12 cells treated with indicated reagents as panel **d**, probed with antibodies against distinct respiratory complexes. **f** SDS-PAGE of whole cell lysates from C2C12 cells with treatment as panel **d**. IB was probed with indicated antibodies. β-Actin was analyzed as an internal control. **g**, **h** Oxygen consumption rate (OCR) measurement of C2C12 cells treated with BAY61-3606 (**g**) or GSK143 (**h**) (1 μM each) and DMSO for 24 h. Data are presented as means ± SE from three biologically independent experiments. Basal respiration, ATP synthesis component of respiration, and maximal respiration were calculated as described in "Methods". FCCP, carbonyl cyanide 4-(trifluoromethoxy) phenylhydrazone. *$P < 0.05$; **$P < 0.01$; unpaired two-sided Student's $t$ test. Source data are provided as a Source Data file.

---

inhibitor-treated mice (Supplementary Fig. 6k, l). These results suggest that SYK inhibitors enhance physical performance by improving the quality of skeletal muscles rather than by increasing muscle mass.

## Discussion

In this study, we established a FRET-based platform that quantifies the interactions between mitochondrial respiratory complexes I and IV, which approximates MRC supercomplex formation in live cells. This platform can be applied to medium-throughput screens as we successfully performed a screen of 1280 pharmacologically active compounds and identified SYK inhibitors as enhancers of MRC supercomplex formation. While we set 40 μM as a threshold concentration of compounds in the primary screening, a possibility

remains that we identified more compounds that promote supercomplex formation at lower concentrations. Our FRET-based assay is unique in that MRC supercomplex abundance can be monitored in live cells while biochemical methods such as BN-PAGE result in the destruction of cells. However, our FRET-based method requires transfection of exogenous sensor genes which are not required for BN-PAGE. Therefore, our FRET-based assay complements BN-PAGE analysis by providing a higher-throughput evaluation of MRC supercomplex formation, which can then be evaluated using BN-PAGE.

Another caveat to our FRET-based assay is that we are approximating the supercomplex formation between complex I and complex IV monomers with a complex III dimer (I/III$_2$/IV) by monitoring the interaction between complex I (NDUFB8) and complex IV (COX8A).

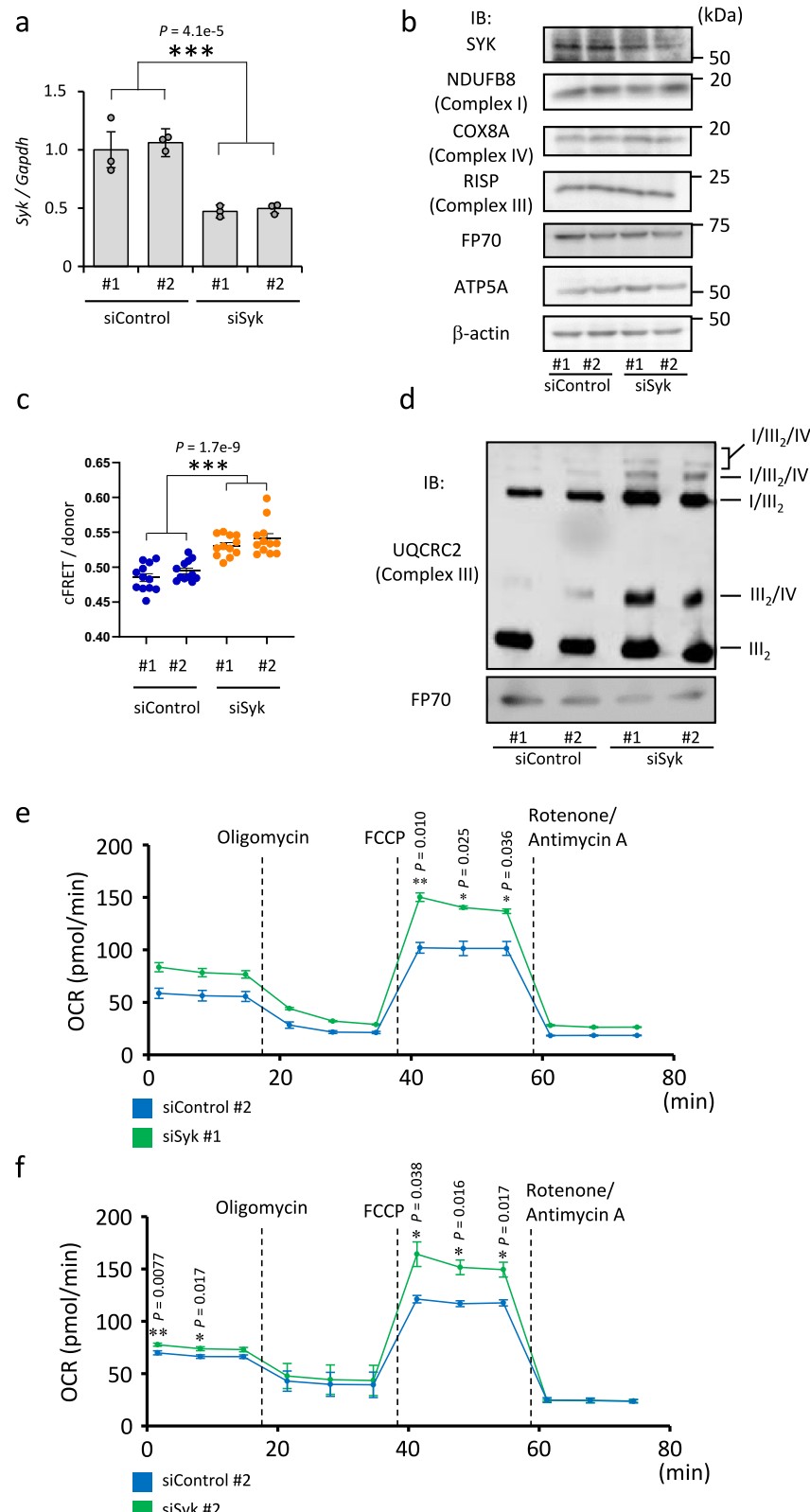

However, there are other MRC supercomplexes which are not detected with our approach. For example, supercomplex assembly factor COX7RP was previously shown to also enhance the abundance of complexes III and IV (III$_2$/IV) supercomplex[6] aside from I/III$_2$/IV supercomplex. Consistent to this role, we also observed decreased abundance of III$_2$/IV supercomplex upon COX7RP silencing using BN-PAGE. Therefore, our FRET-based method identifies a subset of MRC supercomplexes while BN-PAGE can monitor all forms of MRC supercomplexes.

Our assay in its current format is most appropriate for a follow-up assay in large-scale screening campaigns as it uses live cells in 96-well format with commercial image analysis software. To convert our assay into a high-throughput assay capable of processing 10,000 compounds per day (about 30 plates of 384-well plates), we would need to

**Fig. 5 | Inhibition of *Syk* expression increases mitochondrial respiratory chain supercomplex assembly in C2C12 myoblastic cells. a** Knockdown of *Syk* expression in C2C12 cells transfected with its specific siRNAs (siSyk #1 and #2) evaluated by qRT-PCR. Indicated siRNAs (100 pM each) were used. Data are presented as means ± SE of three biologically independent samples. ***$P < 0.001$; two-way ANOVA. **b** Immunoblotting (IB) for mitochondrial protein expression in C2C12 cells transfected with indicated siRNAs (100 pM each), probed with antibodies against distinct respiratory complexes. β-Actin was analyzed as an internal control. **c** Imaging cytometer-based quantification of cFRET/donor ratios for C2C12 cells stably co-expressing NDUFB8-AcGFP and COX8A-DsRed-Monomer transfected with indicated siRNAs. Data are presented as means ± SE from 12 wells. ***$P < 0.001$;

two-way ANOVA. **d** BN-PAGE of mitochondrial proteins from C2C12 cells treated with indicated siRNAs (100 pM each). Positions corresponding to indicated mitochondrial supercomplex and dimerized complex III (III$_2$) are indicated. Immunoblot (IB) was probed with anti-UQCRC2. FP70 was analyzed as an internal control. **e, f** Oxygen consumption rate (OCR) measurement of C2C12 cells transfected with indicated siRNAs for 48 h. Data are presented as means ± SE from three biologically independent experiments. Basal respiration, ATP synthesis component of respiration, and maximal respiration were calculated as described in "Methods". FCCP, carbonyl cyanide 4-(trifluoromethoxy) phenylhydrazone. *$P < 0.05$; **$P < 0.01$; unpaired two-sided Student's *t* test. Source data are provided as a Source Data file.

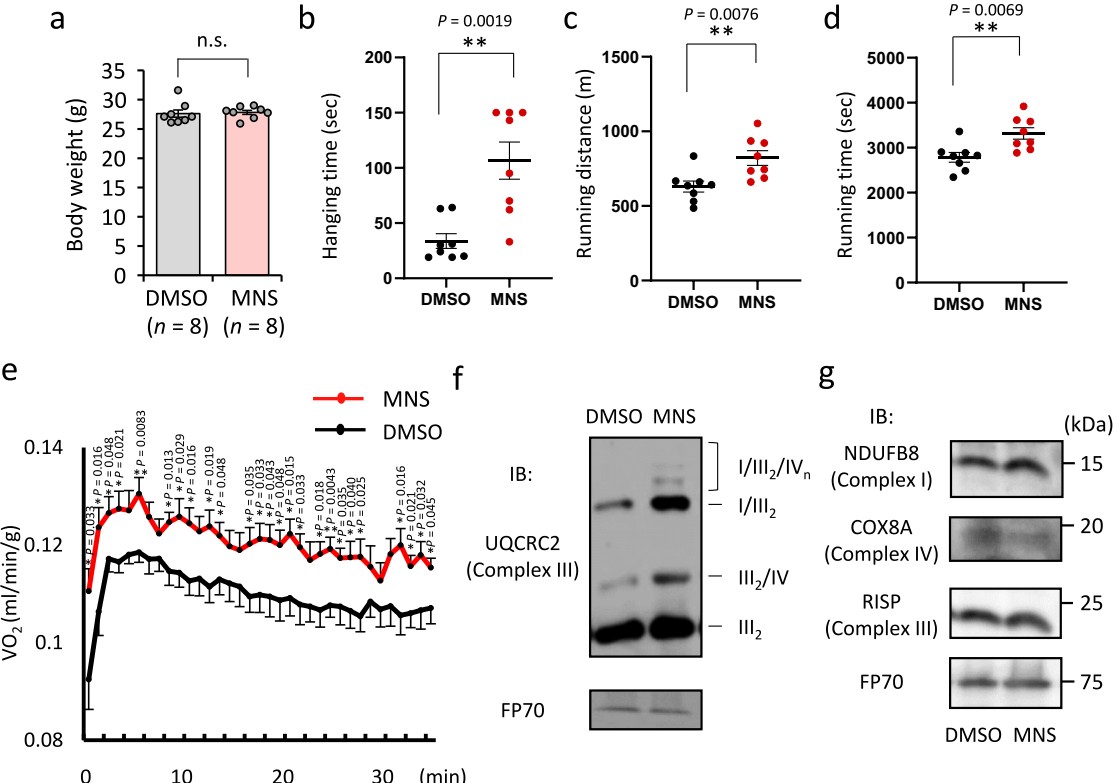

**Fig. 6 | Increased exercise performance in mice treated with MNS. a** Body weights of mice after intraperitoneal injection of MNS (4 mg/kg) or DMSO twice a week for 5 weeks. Data are presented as means ± SE ($n = 8$ biologically independent animals). n.s. not significant; two-sided Mann–Whitney *U* test. **b**, Results of wire hanging test after injection of MNS or DMSO for 3 weeks. Data are presented as means ± SE ($n = 8$ biologically independent animals). **$P < 0.01$; two-sided Mann–Whitney *U* test. **c**, **d** Results of forced treadmill exercise test after injection of MNS or DMSO for 2 weeks. Data are presented as means ± SE ($n = 8$ biologically independent animals). **$P < 0.01$; unpaired two-sided Student's *t* test. **e** O$_2$ consumption (VO$_2$) during forced treadmill exercise test after injection of MNS or DMSO for 4 weeks. Data are presented as means ± SE ($n = 8$ biologically independent animals). *$P < 0.05$; **$P < 0.01$; unpaired two-sided Student's *t* test.

**f** Mitochondrial proteins of quadriceps femoris muscle from DMSO- or MNS-treated mice were solubilized and subjected to BN-PAGE. Positions corresponding to mitochondrial supercomplexes I/III$_2$/IV$_n$, I/III$_2$, III$_2$/IV, and dimerized complex III (III$_2$) are indicated. Western blot analysis was performed with antibody for UQCRC2 of complex III. FP70 protein was blotted as an internal control. IB immunoblot. **g** Mitochondrial proteins of quadriceps femoris muscle from DMSO- or MNS-treated mice were solubilized and subjected to SDS-PAGE. Western blot analysis was performed with antibodies for NDUFB8 of complex I, COX8A of complex IV, and RISP of complex III. FP70 protein was blotted as an internal control. For **f** and **g**, experiments were repeated twice and the results of one experiment are shown. Source data are provided as a Source Data file.

improve the speed of our assay by converting 96-well format into 384-well format, running at least one dedicated automated microscope (for two ROI per well, three channels per well, 2 s per image) with robotic loader for full 24 h, and performing image analysis in parallel computing environment such as using Cell Profiler[54]. Based on the results of Supplementary Fig. 2c, we estimated that the Z'-factor of this assay was -0.200. The Z'-factor in our assay is currently below the expected value in industry-scale high-throughput screens, where 0.500 is the minimal requirement[55]. We expect our Z'-factor to improve by automating cell handling steps in our assay such as cell seeding and

compound addition which we performed manually. Hits from such screen can be followed up by measurement of maximal respiration rate using Flux Analyzer XFe96 (Agilent) in 96-well format, followed by gold-standard BN-PAGE analysis. Alternatively, our FRET-based assay of MRC supercomplex formation in its current 96-well plate form can be run in parallel to Flux Analyzer XFe96 as a secondary or tertiary screens in large-scale screening campaigns.

In this study, we showed that β-lapachone can promote MRC supercomplex assembly in C2C12 myoblastic cells by our FRET-based assay. Given that β-lapachone is an activator for mitochondrial

biogenesis as reported in the literature, the enhancement of super-complex formation may result from the increase in complexes. For example, β-lapachone has been reported to activate peroxisome proliferator-activated receptor gamma coactivator (PGC)-1α, which is a transcriptional coactivator important for mitochondrial biogenesis and type I muscle fiber physiology[56–59], and increases the levels of both MRC complex proteins and supercomplexes[60]. By applying, our FRET-based medium-throughput assay to a compound library screen, we identified a SYK inhibitor MNS as a compound that promotes MRC supercomplex formation. Furthermore, we demonstrated that other SYK inhibitors and siRNA-mediated SYK inhibition enhanced the formation of MRC supercomplex. These pharmacological and genetic results confirm that SYK is a novel suppressor of MRC supercomplex formation. The OCR measurements in C2C12 cells treated with SYK inhibitors and siSyk showed that SYK inhibitors and siSyk enhanced the maximal respiration. The higher OCR in maximal respiration results in the enhanced mitochondrial spare respiratory capacity (SRC). SRC is considered as an important parameter for the adaptation to stress condition, and SRC deficiency is speculated to be associated with neurodegenerative diseases and cardiac diseases[61–63]. Thus, SYK inhibition may affect muscle function partly by promoting the stress adaptive capacity of the mitochondria. Importantly, MNS and BAY61-3606 also increased both the basal and ATP synthesis component of respiration (oligomycin-suppressed OCR), suggesting that additional targets of MNS and BAY61-3606 may be responsible for this additional effect on ATP synthesis. Moreover, we have checked the level of an ATP synthase protein, ATP5A in C2C12 cells treated with SYK inhibitors and siSyk, and found that SYK inhibition did not markedly affect the expression of ATP5A. This result suggests that the difference in the effects of SYK inhibitors on mitochondrial respiration may be independent of ATP synthase biosynthesis. For better understanding of these points, it will be useful to reveal the mechanism by which SYK regulates MRC supercomplex formation.

SYK is a member of the SYK/ZAP-70 family of non-receptor protein tyrosine kinases. The expression of SYK is most frequently found in hematopoietic tissues and is known to play an essential role in the regulation of immune responses and inflammation[64]. SYK-mediated pathways are also involved in cell proliferation, vascular development, and cell adhesion[65]. SYK is phosphorylated by upstream SRC family kinases or autophosphorylated, leading to phosphorylation of SYK targets at tyrosine residues including phosphoinositide 3-kinase (PI3K) regulatory subunit p85[66] and nuclear factor κB (NFκB) inhibitor IκBα[67]. Some of these downstream targets of SYK are involved in cellular metabolism. For example, in CD103 + dendritic cells, activation of IgA receptor Fc alpha receptor I (FcαRI) promotes glycolysis through the activation of SYK and its downstream target PI3K. PI3K is coupled to TBK1 (TANK binding kinase 1) -IKKε (inhibitor of nuclear factor κ B kinase subunit ε)-dependent pathway, which enhances glycolysis to support fatty acid synthesis and enhance endoplasmic reticulum expansion to increase gene translation[68]. While Syk has been shown to play a role in glycolysis activation such as through the activation of PI3K signaling, we showed that the treatment of SYK inhibitors and siRNAs against *Syk* did not significantly change ECARs in C2C12 myoblastic cells. We thus consider that the SYK inhibition-dependent enhancement of mitochondrial respiratory activities is not primarily the counterreaction by the repression of glycolytic pathways via SYK inhibition in our myoblastic cell model.

In addition to metabolic switch mediated by SYK and its downstream targets, SYK may phosphorylate MRC proteins to modulate their activities. For example, tyrosine phosphorylated cytochrome *c*[69] and cytochrome *c* oxidase subunit 1 (COX1)[70] proteins exhibit reduced mitochondrial respiratory activities. Therefore, SYK inhibition may increase OXPHOS activities and oxygen consumption rate by preventing tyrosine phosphorylation in cytochrome *c* and COX1. Although these mechanisms may partly explain for enhanced oxygen

consumption rate, it remains unclear whether any of these contribute to enhanced MRC supercomplex formation or act independently of supercomplex assembly.

One mechanism by which SYK modulates MRC supercomplex formation is through mitochondrial cristae remodeling. Optic atrophy 1 (OPA1), a protein involved in mitochondrial fusion, also promotes formation of tight cristae junction independent of mitochondrial fusion[71,72]. OPA1 was shown to play an essential role in MRC supercomplex assembly and stability and mitochondrial respiratory efficiency[73]. Therefore, SYK inhibitor may enhance MRC supercomplex formation through cristae remodeling. Another mechanism by which SYK inhibitor could enhance MRC supercomplex formation is through upregulation of supercomplex assembly factors such as COX7RP. We previously reported that overexpression of COX7RP in mouse skeletal muscle promoted assembly of MRC supercomplex and enhanced exercise performance[22]. In addition to COX7RP, other assembly factors have also been reported[34]. Given the multiple downstream targets of SYK, multiple mechanisms may likely be involved.

Although SYK is expressed in multiple tissues, it is unclear whether SYK inhibitors promote MRC supercomplex formation throughout the body and further studies are required to clarify it. During the preparation of this manuscript, Bennet et al.[74] reported that they established another medium-throughput quantification method of respirasome assembly based on a nanoluciferase (NanoLuc) binary technology (NanoBiT)[75]. They performed a medium-throughput chemical screen based on this method using U2OS osteosarcoma cells and revealed that the treatment of dihydroorotate dehydrogenase (DHODH) inhibitors promotes the formation of MRC supercomplex[74]. Meanwhile, in their screen, a SYK inhibitor R406 only weakly increased the luminescent signals[74]. In their technique, complexes III and IV are labeled with NanoLuc subunits, whereas in our FRET-based assay, complexes I and IV are labeled with fluorescent proteins to specifically detect respirasome assembly. This difference in approach may be reflected in the differences in screening results. The differences in screening results could also be attributed to the different cell types used. In our study, MRC supercomplex formation is analyzed in C2C12 myoblastic cells in the context of potential application in sarcopenia treatment, which points to a potential role of SYK in MRC supercomplex formation in muscle tissue.

In vivo, the administration of SYK inhibitors did not cause apparent adverse effects including body weight loss. The muscle weights of the SYK inhibitor-treated mice were not significantly altered compared to the corresponding control group. This suggests that SYK inhibitors enhanced muscle function by increasing muscle quality rather than exerting an anabolic effect. SYK inhibitors may be applicable in the prevention and/or treatment of diseases or conditions in which skeletal muscle weakness is involved, such as sarcopenia, mitochondrial diseases, and muscular dystrophy. In addition, enhancing mitochondrial function can be applied to the treatment of diseases or conditions, including heart failure[76], neurodegenerative disorders[77], and metabolic disorders, such as diabetes[78]. Interestingly, SYK promotes actin de-polymerization via phospholipase Cγ2 activation in the process of complement-mediated phagocytosis and promotes phagosome-lysosome fusion through the disappearance of F-actin structure surrounding the phagosome[79]. Therefore, SYK inhibitors may also directly modulate actin polymerization in skeletal muscles, which could contribute to enhanced exercise performance. Notably, DBA/2CrSlc used in the present study is a wild-type strain of DBA/2 line, which possesses 12 amino acids deletion in the proline-rich region of *Ltbp4* gene. This mutant LTBP4 protein exhibits reduced activity to sequester TGF-β, leading to an enhanced activity of the cytokine. This genetic background is relevant to muscular dystrophy susceptibility, thus the introduction of null allele of *dystrophin-associated protein γ-sarcoglycan* (*Sgcg*) into the DBA/2 background exhibited a more severe muscular dystrophy phenotype than the 129T

strain[80]. Intriguingly, SYK inhibition has been shown to decrease the expression of TGF-β in skin and lung and to repress fibrosis in these tissues[81]. While we used only DBA/2CrSlc strain to evaluate the effects of SYK inhibitors on muscle activity, we consider that this strain may be a suitable model for evaluating the impact of SYK inhibitors in muscles because the strain background primarily exhibits susceptibility to muscular dystrophy and fibrosis. A recent study also revealed that anti-LTBP4 treatment reduced muscle fibrosis and enhanced muscle force production[82], thus, further study will elucidate whether the effects of SYK inhibitors on muscle activity can be modulated by LTBP4 expression or TGF-β signaling. Considering that the muscle fibrosis and impaired regenerative satellite cell capacities are often observed in aged patients, it is tempting to speculate that SYK inhibition can be more beneficial for patients with the enhanced TGF-β activity in muscles. For clinical application of SYK inhibitors or hits from future screens, it will be important to first test our hits and its structural analogs in human cells such as primary myoblasts differentiated into myotubes or in induced pluripotent stem cell (iPSC)-derived skeletal muscle given that our screening and follow-up in vivo work were all based on mouse models. For SYK inhibitors, we would also need to establish whether kinases other than SYK are involved in therapeutic effect in vivo using kinome profiling of structural analogs. We would also need to perform extensive pharmacokinetic and pharmacodynamic analysis across tissues in animal models to examine dosage and frequency of compound administration required for each target tissue. Any long-term negative effect of Syk inhibitors on tissue functions needs to be carefully investigated whether these compounds can be clinically applied to muscle disorders with reduced MRC supercomplex formation. For SYK inhibitors, existing data in human clinical trials can be used to help design human studies.

In conclusion, we demonstrated that our FRET-based assay is applicable to a high-throughput screen to evaluate MRC supercomplex assembly. From a screen of the pharmacologically active chemical library, we identified SYK inhibitors as potential therapeutics that promote supercomplex assembly and mitochondrial respiratory function. In vivo, we showed that SYK inhibitors improved exercise performance without any apparent adverse effects. Our study provides insights into the regulatory mechanism of MRC supercomplex formation and would be beneficial in clinical application settings, such as in the development of effective prevention or treatment for conditions involving muscle weakness.

## Methods

### Antibodies and reagents

For antibodies used in Western blotting (WB), mouse monoclonal NDUFB8 antibody (ab110242, 1:2000 dilution), rabbit monoclonal NDUFB8 antibody (ab192878, 1:1000 dilution), mouse monoclonal RISP antibody (ab14746, 1:5000 dilution), rabbit monoclonal ATP5A antibody (ab176569, 1:1000 dilution) and mouse monoclonal UQCRC2 antibody (ab14745, 1:1000 dilution) were purchased from Abcam (Cambridge, UK). Mouse monoclonal FP70 antibody (clone: 2E3GC12FB2AE2, 1:2000 dilution) was purchased from Invitrogen (Waltham, MA, USA). Rabbit polyclonal COX8A antibody (15368-1-AP, 1:1000 dilution) was purchased from Proteintech (Rosemont, IL, USA). Rabbit monoclonal SYK antibody (clone: D3Z1E, 1:1000 dilution) was obtained from Cell Signaling Technology (Beverly, MA, USA). Mouse monoclonal β-actin antibody (clone: AC-74, 1:5000 dilution) was purchased from Sigma (St. Louis, MO, USA). Rabbit polyclonal COX7RP antibody (1:3000 dilution) was raised in our laboratory[22]. Briefly, rabbits were immunized with a peptide comprising 14 amino acid residues of the C-terminal region of the human COX7RP protein. The serum was affinity-purified using Affigel 10 (Bio-Rad, Hercules, CA, USA), according to the manufacturer's instructions. Library of pharmacologically active compounds (LOPAC 1280), 3,4-methylenedioxy-β-nitrostyrene (MNS) and 3,4-dihydro-2,2-dimethyl-2H-naphtho[1,2-b]pyran-5,6-dione (β-lapachone) were purchased from Sigma. 2-(7-(3,4-dimethoxyphenyl)-imidazo[1,2-c]pyrimidin-5-ylamino)-nicotinamide dihydrochloride (BAY61-3606) was obtained from AdooQ Bioscience (Irvine, CA, USA). 2-[[(3 R,4 R)-3-aminotetrahydro-2H-pyran-4-yl]amino]-4-[(4-methylphenyl)amino]-5-pyrimidinecarboxamide dihydrochloride (GSK143) was purchased from Tocris (Ellisville, MO, USA).

### Cell culture

The murine myoblastic cell line, C2C12, was obtained from ATCC (catalog number: CRL-1772, Manassas, VA, USA). C2C12 myoblastic cells were cultured in Dulbecco's modified Eagle's medium (DMEM) supplemented with 10% fetal bovine serum (FBS) and 1% penicillin−streptomycin (FUJIFILM Wako Pure Chemical Corporation, Osaka, Japan) at 37 °C with 5% $CO_2$. DMEM was purchased from Sigma-Aldrich Japan (Tokyo, Japan). To establish stable transfectants, C2C12 clones were selected using G418 (Sigma) at a concentration of 800 μg/mL.

### Plasmid construction and transfection

A plasmid encoding ubiquinol-cytochrome c reductase, 6.4 kDa subunit (UQCR11) was fused with AcGFP as previously described[83]. A plasmid encoding NADH-ubiquinone oxidoreductase subunit B8 (NDUFB8) fused with AcGFP was constructed by subcloning human NDUFB8 into the mammalian expression vector pAcGFP1-N1 (Clontech, Palo Alto, CA, USA). The coding region of NDUFB8 was amplified from the cDNA of the human breast cancer cell line MCF7 using the following primer set:

*NDUFB8*, forward: 5′-CCCGAGCTCGCCATGGGCGCGGTGGC-CAGGGCC-3′

*NDUFB8*, reverse: 5′-GGGGTACCCCGATCTCATAGTGAACCACCC GC-3′.

Plasmids encoding cytochrome c oxidase subunit 8 A (COX8A) or ATP synthase F1 subunit gamma (ATP5F1c) fused with DsRed-Monomer were constructed by changing the fluorophores from the previously constructed plasmids encoding proteins fused with AcGFP[83]. The coding regions of the proteins were subcloned into the mammalian expression vector pDsRed-Monomer-N1 (Clontech). The transfection of expression vectors was performed 24 h after seeding the cells using FuGENE HD (Promega, Madison, WI, USA), according to the manufacturer's instructions.

### Visualization of fluorescent images

Fluorescent images of the cells expressing proteins fused with fluorophores were visualized with confocal fluorescence microscopy, Fluoview 10i (Olympus, Tokyo, Japan) or TCS SP8 (Leica Microsystems, Wetzlar, Germany). In the experiments with Fluoview 10i, AcGFP was excited with a laser wavelength of 489 nm, and emission was detected at 510 nm. The DsRed-Monomer was excited with a laser wavelength of 580 nm, and emission was detected at 610 nm. Mitochondria were visualized using MitoTracker Deep Red FM (Invitrogen) with excitation at 635 nm and emission at 660−760 nm. In the experiments using TCS SP8, AcGFP was visualized with excitation at 488 nm and emission at 493−553 nm. DsRed-Monomer was visualized with excitation at 558 nm and emission at 566−610 nm. Fluoview 10i was used to visualize the fluorescence of cells fixed with 4% paraformaldehyde. TCS SP8 was used to visualize live cells by obtaining images at a speed of 0.002 s per frame. Experiments evaluating the FRET signals were performed with AcGFP as a donor fluorophore and DsRed-Monomer as an acceptor fluorophore. Images showing FRET efficiency were reconstructed using software attached to each fluorescence microscope (Fluoview 10i FRET package, FRET SE-Leica SP8)[84,85].

### Acceptor photobleaching

Acceptor photobleaching was performed according to a previously described method[86] using TCS SP8. Fluorescence signals of both AcGFP and DsRed-Monomer were obtained from C2C12 myoblastic

cells co-expressing AcGFP and DsRed-Monomer at several regions of interest (ROIs). Then, these ROIs were photobleached for 3 min with the maximum energy at 558 nm (excitation wavelength of DsRed-Monomer). Immediately after photobleaching, fluorescent signals of both AcGFP and DsRed-Monomer were obtained from the same ROIs.

## Small interfering RNA transfection
Knockdown of COX7RP and SYK expression was performed by small interfering RNA (siRNA) transfection. Two specific siRNAs targeting each gene and two control siRNAs that did not target human transcripts (siControl #1 and siControl #2) were purchased from RNAi Inc. (Tokyo, Japan). siControl #1 is an siRNA targeting firefly luciferase, and siControl #2 is an siRNA without specific targets. These siRNAs were transfected into C2C12 myoblastic cells at the time of seeding by reverse transfection using Lipofectamine RNAiMAX (Invitrogen) according to the manufacturer's instructions at the indicated concentrations. The siRNA sequences were as follows:

siControl #1
Sense: 5′-GUGGAUUUCGAGUCGUCUUAA-3′
Anti-sense: 5′-AAGACGACUCGAAAUCCACAU-3′
siControl #2
Sense: 5′-GUACCGCACGUCAUUCGUAUC-3′
Anti-sense: 5′-UACGAAUGACGUGCGGUACGU-3′
siCox7rp #1
Sense: 5′-CUGUGGCUUUACGUUAUGAUU-3′
Anti-sense: 5′-UCAUAACGUAAAGCCACAGCA-3′
siCox7rp #2
Sense: 5′-GGCUUUACGUUAUGAUUGACC-3′
Anti-sense: 5′-UCAAUCAUAACGUAAAGCCAC-3′
siSyk #1
Sense: 5′-AAUGAAUUCAACAUACAGGGA-3′
Anti-sense: 5′-CCUGUAUGUUGAAUUCAUUGA-3′
siSyk #2
Sense: 5′-UUAAUCUUGACAGUAAGACAC-3′
Anti-sense: 5′-GUCUUACUGUCAAGAUUAAUU-3′.

## Quantitative reverse-transcription polymerase chain reaction
Quantitative reverse-transcription polymerase chain reaction (qRT-PCR) was performed as previously described[87], with some modifications. Briefly, total RNA was extracted using Sepasol-RNA I Super G (Nacalai Tesque, Kyoto, Japan), followed by cDNA synthesis using PrimeScript (Takara, Kyoto, Japan). The cDNA was subjected to a real-time polymerase chain reaction (PCR) using the Fast 7500 real-time PCR system or StepOne real-time PCR system (Applied Biosystems, Foster City, CA, USA) based on the detection of SYBR Green fluorescence (Kapa Biosystems, Woburn, MA, USA). The mRNA expression levels were normalized to the expression level of *Gapdh* using the 2-ΔΔCT method[88]. The primer sequences are as follows:

*Gapdh*, forward: 5′-GGTGGTCTCCTCTGACTTCAACA-3′
*Gapdh*, reverse: 5′- GTGGTCGTTGAGGGCAATG-3′
*Cox7rp*, forward: 5′-TACAAGTTTAGCAGTTTCACGCAG-3′
*Cox7rp*, reverse: 5′-AGGTCAGTTTGGTTGGTGTGG-3′
*Syk*, forward: 5′-GGAAGAGAGCAACTTTGTGC-3′
*Syk*, reverse: 5′-GTCTGGGCCTTGTAGTAGTT-3′.

## Preparation of the mitochondrial fraction
To extract the mitochondrial fraction, skeletal muscle or C2C12 myoblastic cells were homogenized in homogenization buffer containing 10 mM HEPES-KOH (pH 7.4, 0.22 M mannitol, 0.07 M sucrose, and 0.1 mM EDTA). Cell extracts were centrifuged at $500 \times g$ for 15 min, and the supernatant was precipitated by further centrifugation at $10,000 \times g$ for 15 min. The precipitant was resuspended in homogenization buffer and precipitated again by centrifugation at $10,000 \times g$ for 15 min. The precipitant was suspended in

homogenization buffer and used as the mitochondrial fraction for further analysis.

## Sodium dodecyl sulfate-polyacrylamide gel electrophoresis (SDS-PAGE) and western blot analysis
The protein concentration of the mitochondrial fraction was measured using the BCA Protein Assay (Pierce, Rockford, IL, USA). The samples were mixed with 4× sample buffer (0.4 M Tris−HCl, pH 6.8, 8% SDS, 0.4 M dithiothreitol, 40% glycerol, and 0.04% bromophenol blue) and boiled for 5 min. The samples were separated using SDS-PAGE and then transferred to polyvinylidene difluoride (PVDF) membranes (Millipore, Darmstadt, Germany or Pall Corporation, Port Washington, NY, USA). The membranes were blocked in Bullet Blocking One (Nacalai Tesque) for 5 min. The membranes were incubated with primary antibodies, followed by incubation with horseradish peroxidase (HRP)-conjugated secondary antibody (NA931V or NA934V, 1:2500 dilution for WB, GE Healthcare, Buckinghamshire, UK). The bound antibodies were visualized using Chemilumi-One Ultra (Nacalai Tesque). For re-probing, the antibodies were stripped using WB stripping solution (FUJIFILM Wako Pure Chemical Corporation). Uncropped scans of the immunoblots were shown in Source Data and Supplementary Fig. 7.

## Blue native PAGE
The mitochondrial fraction (50 μg mitochondrial protein) was suspended in 15 μL of a buffer containing 50 mM HEPES-KOH (pH 7.4), 150 mM potassium acetate, and 10% glycerol. Digitonin (10%) was added to solubilize the mitochondria at a digitonin/protein ratio of 8 g/g. After incubation on ice for 30 min, the solubilized proteins were obtained as the supernatant fraction by centrifugation for 15 min at $20,000 \times g$ at 4 °C. A stacking gel (4%) and separating gels with stepwise 8, 9, 10, and 11% were cast, and the solubilized proteins were electrophoresed according to a previously described method[22,89,90]. The gels were subjected to western blot analysis.

## Quantification of FRET signal
FRET signals in the cells co-expressing AcGFP and DsRed-Monomer in a 96-well plate were quantified using IN Cell Analyzer 6000 (GE Healthcare, Chicago, IL, USA) and IN Cell Investigator image analysis software version 1.6.2 (GE Healthcare). The cells were treated with compounds 24 h before examination or transfected with siRNAs 48 h before the examination. Fluorescence signals of AcGFP (excitation 488 nm, emission 522 nm) and DsRed-Monomer (excitation 561 nm, emission 605 nm) and raw FRET signal (excitation 488 nm, emission 605 nm) were obtained in 12 ROIs per well. To evaluate the bleed-through of AcGFP signal to the raw FRET signal, C2C12 myoblastic cells stably expressing only NDUFB8-AcGFP were excited with a laser wavelength of 488 nm, and emission was detected at 605 nm. Based on this experiment, the bleed-through coefficient (α) in the following equation was determined. Next, to evaluate the cross-excitation of DsRed-Monomer at 488 nm, C2C12 myoblastic cells stably expressing only COX8A-DsRed-Monomer were excited with a laser wavelength of 488 nm, and emission was detected at 605 nm. Based on this experiment, the cross-excitation coefficient (β) in the following equation was determined. The corrected FRET (cFRET) values were calculated as follows: cFRET = (raw FRET signal intensity) − α × (donor signal intensity) − β × (acceptor signal intensity)[84]. To evaluate the effects of chemicals or siRNAs on FRET signals, the cFRET values were divided by donor signal intensities (cFRET/donor)[43–47].

## Measurements of cellular oxygen consumption
The oxygen consumption rate (OCR) and extracellular acidification rate (ECAR) were measured with an XF Extracellular Flux Analyzer (Seahorse Biosciences, Billerica, MA, USA) using the Cell Mito Stress Test Kit (Seahorse Biosciences) as previously described[30]. Cells were seeded ($5 \times 10^3$ cells/well) in an XF Cell Culture Microplate (Seahorse

Biosciences) and cultured overnight before measuring their OCR and ECAR. For the knockdown experiments, the cells were transfected with siRNAs for 48 h, as indicated in the figure legends. One hour before the assay, the culture medium was replaced with XF Base Medium Minimal DMEM containing the substrates (1 mM sodium pyruvate, 2 mM glutamine, and 25 mM glucose) (Seahorse Biosciences), in which SYK inhibitors and siSyk were not added. OCR and ECAR were measured at 37 °C. The baseline (basal) OCR and ECAR were measured in triplicate before and after each sequential injection of oligomycin (1 μM), carbonyl cyanide 4-(trifluoromethoxy) phenylhydrazone (FCCP; 0.5 μM), and rotenone and antimycin A (both 0.5 μM). Basal respiration, ATP synthesis component of respiration, and maximal respiration were calculated as below. Basal respiration was measured as the OCR before oligomycin injection minus the non-mitochondrial respiration. ATP synthesis component of respiration was measured as the OCR prior to oligomycin injection minus the OCR after oligomycin injection. Maximal respiration was measured as the OCR after FCCP addition minus the non-mitochondrial respiration.

### Assessment of functional performance of mice

Seven-week-old male DBA/2CrSlc mice were purchased from Japan SLC, Inc. (Shizuoka, Japan). After acclimatizing to the environment for 1 week, the mice were intraperitoneally administered SYK inhibitors twice a week at the following doses: MNS (4 mg/kg), BAY61-3606 (2 mg/kg), and GSK143 (2 mg/kg). The drugs were diluted tenfold with dimethyl sulfoxide (DMSO) before further dilution with physiological saline at the working concentration. In the control group, the mice were treated with DMSO at a dose of 1.5 mL/kg as a vehicle in 100 μL physiological saline.

The hanging time was measured using a wire hanging chamber (O'Hara & Co. Ltd., Tokyo, Japan). The mice were accustomed to the equipment by subjecting them once prior to the exercise performance test. The hanging time was measured twice and the longer time was adopted for the data. The maximum measuring time was set to 150 s. Running time and distance were measured using a treadmill (Treadmill For Mice Model LE8710M; Panlab Harvard Apparatus, Barcelona, Spain). Before the exercise performance test, the mice were accustomed to the treadmill by subjecting them to the actual test regimen as follows: a 15 cm/s run for the first 10 min, followed by an incremental increase of 2 cm/s at 10 min intervals. Exhaustion was defined as a state in which the mice were unable to avoid repetitive electrical shocks. The respiratory metabolic rate was measured using an individual indirect calorimetric system equipped with an airtight treadmill (Modular Enclosed Telemetric & Metabolic Treadmill 1013M-1; Columbus Instruments, Columbus, OH, USA). The test regimen comprised 4 min at rest inside the chamber, followed by 20 cm/s with no change in the speed for 36 min. During this time, the data for each chamber were collected every minute, with room air used as the reference using which the $O_2$ consumption was calculated.

All mice used in the experiments were maintained in a specific pathogen-free mouse facility of Tokyo Metropolitan Institute of Gerontology at a temperature of 22 ± 2 °C, a relative humidity of 55 ± 5%, and a 12:12 h light:dark cycle (lights on, 08:00 to 20:00), with free access to water and Low Irradiated Diet (CRF-1, LID6, Oriental Yeast Co., Ltd., Tokyo, Japan). The colony was routinely monitored for, and found to be free of, 4 viruses (ectromelia virus, lymphocytic choriomeningitis (LCM) virus, mouse hepatitis virus, Sendai virus), 6 bacterial species (*Corynebacterium kutscheri*, Salmonella spp., *Clostridium piliforme*, *Citrobacter rodentium*, *Pasteurella pneumotropica*, *Mycoplasma pulmonis*), and external and internal parasites. Mice were observed daily and euthanized by cervical dislocation at the end of each experiment. All studies involving animals were reviewed and approved by the Institutional Animal Care and Use Committee of the Tokyo Metropolitan Institute of Gerontology (approval no. 18021). This study complied with ARRIVE guidelines (https://arriveguidelines.org/).

### Transmission electron microscopic examination

The skeletal muscles from drug-treated mice were fixed with 2.5% glutaraldehyde in 0.1 M phosphate buffer (pH 7.4), post-fixed for 1 h with 2% $OsO_4$ dissolved in distilled water, dehydrated in a graded series of ethanol solutions, and embedded in Epon. Ultrathin sections were generated using an ultramicrotome (REICHERT ULTRACUT S, Leica Microsystems) and stained with uranyl acetate and lead citrate for examination under a transmission electron microscope (H-7500; Hitachi High-Technologies, Tokyo, Japan).

### Statistical analysis

Statistical analyses were conducted using Excel Statistics 2010 (add-in software for Microsoft Excel) (SSRI, Tokyo, Japan), GraphPad Prism 8 (GraphPad Software, San Diego, CA, USA), or JMP version 9.0.0 (SAS Institute, Cary, NC, USA). The normality of the experimental data was examined by Shapiro–Wilk test. The $EC_{50}$ of the chemicals was calculated by fitting the dose-response curve to a four-parameter logistic equation: $(y = d + (a - d)/[1 + (x/c)^b])$ using ImageJ software (https://imagej.nih.gov/ij/download.html).

### Reporting summary

Further information on research design is available in the Nature Portfolio Reporting Summary linked to this article.

## Data availability

All the other data supporting the findings of this study are available from the corresponding authors upon request. Source data are provided with this paper.

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

## Acknowledgements

We thank Mr. F. Hasegawa for his technical assistance of the electron microscopic examination. We also thank Dr. I. Ohsawa and Dr. A. Sawano for their valuable technical advice on imaging techniques. This study was supported by grants from the Japan Society for the Promotion of Science (20K21667 and 21H04829 to S.I., 20H03734 to K.H., 20K08916, 20K21636, and 21H02981 to K.I.), Takeda Science Foundation (to S.I. and K.I.), the Vehicle Racing Commemorative Foundation (to K.H.) and Terumo Life Science Foundation (K.A).

## Author contributions

K.A., T.K., and S.I. designed research; A.K., K.A., T.T., and K.I. performed research; T.K. and K.I. contributed new reagents/analytic tools; A.K., K.A., and T.T. analyzed the data; A.K. and K.A. prepared original draft; K.H. and S.I. edited the manuscript. All authors critically reviewed the manuscript.

## Competing interests

The authors declare no competing interests.
