## [Peer Review File · Nature Communications]

FRET-based respirasome assembly screen defines potential therapeutic intervention that improves muscle mitochondrial respiration and exercise performanceREVIEWER COMMENTS

Reviewer #1 (Remarks to the Author):

This is an impressive study that claims to establish feasibility for high-throughput screening to detect compounds that affect mitochondrial function. The paper could be improved by more clear discussion of the next steps required to approach effective drug discovery and development. Most notably, this paper describes a screening using the LOPAC library, a collection of 1280 pharmaceutically active compounds that are used to provide preliminary validation of a developing screening technology. Clearly, this small screen does not establish high probability that the desired drugs will be developed in the near future. Therefore, the paper can be improved by discussing in more quantitative detail what is the potential for substantially expanding the scale of this assay, to a large enough chemical library to be consistent with the industry standard -- at least 50,000 compounds, but more likely 1,000,000 compounds or more. Does this assay have the potential to be scaled up to that level? Does it have the combination of speed (e.g., 10,000 compounds per day) and precision (e.g., assay quality Z-prime-factor greater than 0.5, measured under HTS-compatible conditions) needed for scaling up to a large library? Are the secondary assays also of sufficient speed and precision to evaluate the hits from the primary screen?

A brief discussion of the future testing and development (e.g., in animals, in human cell culture, and in humans), including medicinal chemistry, that would be needed to develop useful drugs.

Once the above questions have been answered more clearly, this paper is likely to be interesting to a wide range of scientists, both in academia and industry.

I would be happy to review a revised manuscript.

Reviewer #2 (Remarks to the Author):

This manuscript by Kobayashi and colleagues uses FRET to characterize mitochondrial respirasome assembly in the context of muscle physiology. The FRET biosensors are shown to be effective in detecting the organization of mitochondrial respiratory chain enzymes and used properly through the manuscript. The major concern of this reviewer is the potential overinterpretation of the results. It is necessary to distinguish whether effects on cell respiration and muscle performance are due to enhanced mitochondrial biogenesis or directly to changes in the organization of the mitochondrial respiratory chain. Specific comments are listed below.

Abstract

1. The abstract states: "The assembly of mitochondrial respiratory complexes into higher-order "supercomplex" structures is nowadays considered as an efficient biological process for energy synthesis. " However, there is significant controversy in the field regarding the physiological significance of the organization of the respiratory chain complex enzymes into supercomplexes.
2. SYK needs to be defined in the abstract.
3. "SYK inhibition in myoblastic cells increases supercomplex assembly and mitochondrial respiration". This sentence is misleading since the paper shows correlation between supercomplex association and mitochondrial respiration but not a cause-effect relationship.

Introduction

1. Lines 73-74. References 3-4 do not show that "Supercomplex assembly enhances respiratory chain activity and mitochondrial respiration while minimizing reactive oxygen species (ROS) production through the spatial restriction of electron carrier diffusion" QAs mentioned above, the field is very controversial, and the introduction should reflect this by citing all literature in a non-biased manner.

1. Line 85. "However, the molecular pathways regulating MRC supercomplex formation are largely unknown, partly owing to time-consuming and complex assays involved." I do not think that time-consuming assays are delaying progress in the field. Perhaps this needs to be rephased.

Results

1. Fragment about β -Lapachone starting in line 161. This compound seems to enhance mitochondrial biogenesis and therefore is not surprising that complexes and supercomplexes are also enhanced. A different issue is whether it directly promotes supercomplex assembly as stated in the text.
2. In general, compounds that enhance mitochondrial biogenesis, enhance the levels of mitochondrial complexes and supercomplexes, and have been also found to improve muscle physiology. The effect is similar to the previously-reported overexpression of PGC1-alpha. This should be mentioned in the text.
3. The data regarding BAY61 and GSK143 is difficult to interpret because figure 4e shows the steady-state levels of respiratory chain proteins "using mitochondrial proteins from C2C12 myoblastic". The WB should be done in whole cell extracts to allow for the evaluation of potential increases in mitochondrial mass.
4. The quality of some WB is not optimal, particularly regarding the loading controls, which surely made it difficult to perform accurate quantifications. Examples are in Fig 4d and Fig 5d.

Reviewer #3 (Remarks to the Author):

This study validates a FRET-based approach of detecting mitochondrial respiratory supercomplex formation in cells. Using this approach, drugs were screened and identified for their ability to modulate supercomplex formation using FRET as a read-out in addition to verification by BN-PAGE. These experiments led to an in vivo treatment approach that demonstrated positive effects of the drugs on locomotor functional tests in mice relative to DMSO injected controls that were associated with BN-PAGE assessed changes in supercomplex formation.

The study is interesting and logical in its progression. Appropriate methodologies have been used overall. There are several uncertainties outlined below. The major areas of uncertainty are whether the DMSO dose injected in vivo was too high (10%) which would require a non-DMSO control group, whether the screening approach based on transfection approaches for FRET are considerably more advantageous than BN-PAGE (more specific elaboration of the benefit of this approach in cells is important), and whether the assumed components of the supercomplex in question were captured in the assay as noted below.

Specific comments:

Line 111-112: A very recent preprint identifies Complexes I, II, III and IV can exist as a megacomplex. While this is not peer reviewed, it introduces uncertainty regarding the assumption of the present paper that I and IV do not bind. Given this assumption guided the interpretation that FRET-based detection of I-IV must mean the supercomplex I-III2-IV were detected, it seems that a need for clarification of the assumption would be beneficial. What confidence can be provided that I-IV detections do not reflect alternative super/mega complexes as identified in this preprint?

<https://www.biorxiv.org/content/10.1101/2022.06.26.497646v1>

Or see this link:

doi: <https://doi.org/10.1101/2022.06.26.497646>

For experiments in Figure 3, was the concentration of DMSO verified to have no effect on the FRET signal? A similar question pertains to the in vivo injections in Figure 6. The methods indicate that mice were injected with 10% DMSO which seems high given lower concentrations are often used in drug studies. Comparison of DMSO treated samples and mice to saline-injections (to control for the stress of injections vs DMSO) would be important at this seemingly high DMSO dose.

Figure 3/Supplemental Table S1: what was the procedure for deciding on the concentration to use for each candidate compound for the initial screen (40uM)? The finding that most of the 222 chemicals with a FRET value above 0.6 caused cell death or change in cell morphology might not exclude their potential role at lower concentrations, for example. Could the authors comment on the possibility that

some of these compounds might still have potential efficacy at doses not tested in this study?

Fig 3 figure legend might list the duration of treatment for MNS

Fig 3f – the ATP synthesis component of the OCR trace could be determined and tested for differences between MNS and control (reduction with oligomycin; the actual absolute loss of OCR with oligomycin rather than the new steady state which is more attributable to leak-induced respiration). This applies to the remaining OCR graphs in subsequent figures as well.

In general, the supercomplex blots (Fig 3D, 4D as examples) do not convey consistency of findings given these are n=1. How reproducible are these findings?

Fig 4g – the finding that GSK143 only increased 'maximal' OCR can be discussed in terms of its utility as a index of maximal flux through the ETS vs the lack of physiological relevance of this measure given it is an uncoupled phenomenon and there was no change in the apparent ATP synthesis component of basal respiration (although this requires the t-test on the absolute drop in OCR with oligomycin). A similar question pertains to Fig 5F.

Fig 6 - <60 seconds cage hang time is quite low for a healthy wildtype mouse, although perhaps this is strain dependent. Some strains can hold on for over 150 seconds which often leads to the requirement to set a maximal time for the test (it seems a similar threshold was set in extended figure 5d for the GSK group). Similar to the question above, did DMSO affect cage hang time? Does this time not seem low?

Conclusion line 369 – the basis for determining effective preventative treatments for muscle weakness could be rephrased to be more speculative and broad as no model of muscle weakness was assessed.

Model: the DBA/2 line of mice is known to have a natural mutation in TGF-beta binding protein 4 leading to elevated activity of TGF-beta. This model has elevated fibrosis and impaired regenerative satellite cell capacities amongst other distinct attributes. Can the authors comment on the impact of this natural mutation on their data? However, I am not sure if DBA/2 is the same strain as DBA/2CrSlc used in this study.

Functional tests: more detail on the cage hang time could be included. Were measures made in duplicate or single tests? Was 150 seconds set as the maximum time as implied in the extended data figure 5?

TEM: how many sections were assessed per sample and how many images per section were quantified? Extended fig 4E gives an 'n' but does not indicate if these are sections or areas within a single section (likely the latter, but how many sections?).

Was normality verified before using parametric tests where appropriate?

OCR: Was the drug present during the OCR assay in the cell experiments, or were the cells pre-treated with the drug without the drug present in the media during the actual assay?

Responses to Reviewers

Reviewer #1 (Remarks to the Author):

Q1: This is an impressive study that claims to establish feasibility for high-throughput screening to detect compounds that affect mitochondrial function. The paper could be improved by more clear discussion of the next steps required to approach effective drug discovery and development. Most notably, this paper describes a screening using the LOPAC library, a collection of 1280 pharmaceutically active compounds that are used to provide preliminary validation of a developing screening technology. Clearly, this small screen does not establish high probability that the desired drugs will be developed in the near future. Therefore, the paper can be improved by discussing in more quantitative detail what is the potential for substantially expanding the scale of this assay, to a large enough chemical library to be consistent with the industry standard -- at least 50,000 compounds, but more likely 1,000,000 compounds or more. Does this assay have the potential to be scaled up to that level? Does it have the combination of speed (e.g., 10,000 compounds per day) and precision (e.g., assay quality Z'-factor greater than 0.5, measured under HTS-compatible conditions) needed for scaling up to a large library? Are the secondary assays also of sufficient speed and precision to evaluate the hits from the primary screen?

Ans.: We appreciate the valuable comments raised by this reviewer.

In terms of the potential to be scaled up our compound screening to the standard industrial level, we understand that our primary FRET-based assay and secondary assays need to be improved for the assay speed and precision. From this point of view, we now define our assay as “medium-throughput” screen instead of “high-throughput” one in the revised text. For our FRET-based assay, the rate limiting step is currently the simultaneous running of image capture by *IN Cell Analyzer 6000* system and image processing and quantification by *IN Cell Investigator*. While it takes ~ 5 hours to be processed for the capture for 12 regions of interest (ROI) using 3 channels (AcGFP, DsRed-Monomer, raw FRET signal) per well in 96-well format, the assay speed may be remarkably improved by modulating several conditions: 384-well application, capture of fewer ROI per well, simultaneous running of multiple automated microscopes using robotic arms, and multiple running of image analysis using parallel computing environment (e.g., CellProfiler platform) for large-scale bioimage processing. The application of semi or full automation of cell manipulation instead of current manual procedure will further shorten the assay time. If so, we assume that the Z'-factor of the FRET-based assay will be increased to reach 0.5, which is currently ~ 0.2 as analyzed in Extended Fig. 2c. Nevertheless, our primary assay may have an

impact on drug screen as medium-throughput screen, because we successfully identified biologically significant hits in the present study.

For secondary assays, OCR measurement by Seahorse Flux Analyzer can be performed in 96-well format, whereas BN-PAGE is manually performed and needs more time for cell and protein manipulation, and electrophoresis. We consider that OCR measurement can be improved if 386-well format is applied in future whereas it is difficult to simplify the procedure of BN-PAGE at this point. Considering the speed and precision of these assays, we expect to apply OCR measurement as the secondary assay and BN-PAGE as the tertiary assay for the compound screen.

We noted these points in the Discussion section of the revised manuscript as below.

“Our assay in its current format is most appropriate for a follow-up assay in large-scale screening campaigns as it uses live cells in 96-well format with commercial image analysis software. To convert our assay into high-throughput assay capable of processing 10,000 compounds per day (about 30 plates of 384-well plates), we would need to improve the speed of our assay by converting 96-well format into 384-well format, running at least one dedicated automated microscope (for 2 ROI per well, 3 channels per well, 2 seconds per image) with robotic loader for full 24 hours, and performing image analysis in parallel computing environment such as using Cell Profiler⁵⁴. Based on the results of Extended Data Figure 2c, we estimated that the Z'-factor of this assay was approximately 0.200. The Z'-factor in our assay is currently below the expected value in industry-scale high-throughput screens, where 0.500 is the minimal requirement⁵⁵. We expect our Z'-factor to improve by automating cell handling steps in our assay such as cell seeding and compound addition which we performed manually. Hits from such screen can be followed up by measurement of maximal respiration rate using Flux Analyzer XFe96 (Agilent) in 96-well format, followed by gold-standard BN-PAGE analysis. Alternatively, our FRET-based assay of MRC supercomplex formation in its current 96-well plate form can be run in parallel to Flux Analyzer XFe96 as a secondary or tertiary screens in large-scale screening campaigns.” (lines 322-338 on pages 13-14 of the revised manuscript)

New ref #54. Stirling, D.R., Swain-Bowden, M.J., Lucas, A.M., Carpenter, A.E., Cimini, B.A. & Goodman, A. CellProfiler 4: improvements in speed, utility and usability. *BMC Bioinformatics* **22**, 433 (2021).

New ref #55. Zhang, J.H., Chung, T.D. & Oldenburg, K.R. A simple statistical parameter for use in evaluation and validation of high throughput screening assays. *J. Biomol. Screen.* **4**, 67-73 (1999).

Q2: A brief discussion of the future Testing and development (e.g., in animals, in human cell culture, and in humans), including medicinal chemistry, that would be needed to develop useful drugs.

Ans.: As pointed out by the reviewer, we have added the following sentences to the manuscript:

For clinical application of SYK inhibitors or hits from future screens, it will be important to first test our hits and its structural analogs in human cells such as primary myoblasts differentiated into myotubes or in induced pluripotent stem cell (iPSC)-derived skeletal muscle given that our screening and follow-up *in vivo* work were all based on mouse models. For SYK inhibitors, we would also need to establish whether kinases other than SYK are involved in therapeutic effect *in vivo* using kinome profiling of structural analogs. We would also need to perform extensive pharmacokinetic and pharmacodynamic analysis across tissues in animal models to examine dosage and frequency of compound administration required for each target tissue. Any long-term negative effect of Syk inhibitors on tissue functions needs to be carefully investigated whether these compounds can be clinically applied to muscle disorders with reduced MRC supercomplex formation. For SYK inhibitors, existing data in human clinical trials can be used to help design human studies.” (lines 445-456 on pages 18-19 of the revised manuscript)

Q3: Once the above questions have been answered more clearly, this paper is likely to be interesting to a wide range of scientists, both in academia and industry. I would be happy to review a revised manuscript.

Ans.: We sincerely appreciate the reviewer’s constructive comments.

Reviewer #2 (Remarks to the Author):

This manuscript by Kobayashi and colleagues uses FRET to characterize mitochondrial respirasome assembly in the context of muscle physiology. The FRET biosensors are shown to be effective in detecting the organization of mitochondrial respiratory chain enzymes and used properly through the manuscript. The major concern of this reviewer is the potential overinterpretation of the results. It is necessary to distinguish whether effects on cell respiration and muscle performance are due to enhanced mitochondrial biogenesis or directly to changes in the organization of the mitochondrial respiratory chain. Specific comments are listed below.

Ans.: We appreciate the valuable comments from this reviewer.

As suggested by the reviewer, we performed WB experiments using whole cell lysates to investigate the effect of SYK inhibitors or siSyk treatment on the expression of respiratory chain complex proteins in C2C12 cells. The results showed that treatment with SYK inhibitors or siSyk did not obviously affect the amount of respiratory chain complex proteins in whole cell lysates, suggesting that SYK inhibition may directly affect MRC supercomplex assembly. The WB data are shown in Figs. 3f, 4f, 5b in the revised manuscript. In addition, we have described the results in the revised manuscript as follows:

“We found that protein levels of NDUFB8 (complex I), RISP (complex III), and COX8A (complex IV) in the mitochondrial fraction and whole cell lysates were not markedly changed in response to MNS (Fig. 3e, f), suggesting that increased abundance of mitochondria, or mitochondrial biogenesis, may not be the primary reason for increased MRC supercomplex formation.” (lines 229-233 on page 10 of the revised manuscript)

“These additional SYK inhibitors also did not obviously increase the protein abundance of respiratory chain subunits in the mitochondrial fraction and whole cell lysates (Fig. 4e, f).” (lines 246-247 on page 11 of the revised manuscript)

“The protein concentrations of respiratory chain subunits in whole cell lysates were not markedly changed by silencing SYK expression (Fig. 5b)” (lines 258-259 on page 11 of the revised manuscript)

The responses to the specific comments are described below.

Abstract

1. The abstract states: “The assembly of mitochondrial respiratory complexes into higher-order “supercomplex” structures is nowadays considered as an efficient biological process for energy synthesis.” However, there is significant controversy in

the field regarding the physiological significance of the organization of the respiratory chain complex enzymes into supercomplexes.

Ans.: As the reviewer pointed out, we understand the controversy over the physiological relevance of MRC supercomplex and modified the description as below.

Old version: “The assembly of mitochondrial respiratory complexes into higher-order “supercomplex” structures is nowadays considered as an efficient biological process for energy synthesis.” (lines 53-54 on page 3 of the original manuscript)

Revised version: “The assembly of mitochondrial respiratory complexes into higher-order “supercomplex” structures has been proposed to be an efficient biological process for energy synthesis, although there is controversy in its physiological relevance.” (lines 52-54 on page 3 of the revised manuscript)

2. *SYK needs to be defined in the abstract.*

Ans.: As suggested by the reviewer, we introduced the full spelling of SYK and the family in which SYK belongs to in the Abstract as follows:

Old version: “FRET-based high-throughput live screen defined that SYK inhibition in myoblastic cells...” (lines 58-59 on page 3 of the original manuscript)

Revised version: “FRET-based live cell screen defined that the inhibition of spleen tyrosine kinase (SYK), a non-receptor protein tyrosine kinase that belongs to the SYK/zeta-chain-associated protein kinase 70 (ZAP-70) family, leads to an increase in supercomplex assembly in myoblastic cells.” (lines 58-61 on page 3 of the revised manuscript)

3. *“SYK inhibition in myoblastic cells increases supercomplex assembly and mitochondrial respiration”. This sentence is misleading since the paper shows correlation between supercomplex association and mitochondrial respiration but not a cause-effect relationship.*

Ans.: As pointed out by the reviewer, we understand that our study did not directly reveal a cause-effect relationship between supercomplex formation and mitochondrial respiration. We thus changed the sentence in lines 58-59 of the original manuscript as follows.

Old version: “FRET-based high-throughput live screen defined that SYK inhibition in myoblastic cells increases supercomplex assembly and mitochondrial respiration.” (lines 58-59 on page 3 of the original manuscript)

Revised version: “FRET-based live cell screen defined that the inhibition of spleen tyrosine kinase (SYK), a non-receptor protein tyrosine kinase that belongs to the SYK/zeta-chain-associated protein kinase 70 (ZAP-70) family, leads to an increase in supercomplex assembly in myoblastic cells. In parallel, SYK inhibition enhanced mitochondrial respiration in the cells.” (lines 58-62 on page 3 of the revised manuscript)

Introduction

1. Lines 73-74. References 3-4 do not show that “Supercomplex assembly enhances respiratory chain activity and mitochondrial respiration while minimizing reactive oxygen species (ROS) production through the spatial restriction of electron carrier diffusion” As mentioned above, the field is very controversial, and the introduction should reflect this by citing all literature in a non-biased manner.

Ans.: As suggested, we cited more references and mentioned the controversy regarding the physiological role of MRC supercomplex in the Introduction section of the revised manuscript as below.

Old version: “Supercomplex assembly enhances respiratory chain activity and mitochondrial respiration while minimizing reactive oxygen species (ROS) production through the spatial restriction of electron carrier diffusion.” (lines 72-74 on page 4 of the original manuscript)

Revised version: “The physiological significance of the MRC supercomplexes are still in debate. Some reports suggest that the MRC supercomplex assembly enhances efficacy of respiratory chain reaction by shortening the distance between respiratory complexes^{3,4}, by modulating the assembly and stability of complex I⁵, or by changing the structural surface for quinone-binding sites⁶. In addition, substrate channeling and sequestration of quinone pool are proposed^{1,7-10} as important changes induced by MRC supercomplexes. However, recent biochemical and structural data question these hypotheses¹¹⁻¹⁴. MRC supercomplex assembly also minimizes excessive ROS production^{8,15,16} although this point still remains controversial. Furthermore, the “plasticity model” of MRC has been proposed in which the MRC supercomplexes contribute to metabolic adaptation by rearranging themselves in response to changes in metabolic source of electrons¹⁶⁻²⁰. All of these point to the importance of developing new tools for studying and manipulating MRC supercomplexes.” (lines 74-85 on page 4 of the revised manuscript)

New ref #3. Stuchebrukhov, A., Schäfer, J., Berg, J. & Brzezinski, P. Kinetic advantage of forming respiratory supercomplexes. *Biochim. Biophys. Acta Bioenerg.* **1861**, 148193 (2020).

New ref #4. Berndtsson, J. et al. M. Respiratory supercomplexes enhance electron transport by decreasing cytochrome c diffusion distance. *EMBO Rep.* **21**, e51015 (2020).

- New ref #5. Enríquez, J.A. Supramolecular Organization of Respiratory Complexes. *Annu. Rev. Physiol.* **78**, 533-61 (2016).
- New ref #6. Vercellino, I. & Sazanov, L.A. Structure and assembly of the mammalian mitochondrial supercomplex CIII₂CIV. *Nature* **598**, 364-367 (2021).
- New ref #7. Lapuente-Brun, E. et al. Supercomplex assembly determines electron flux in the mitochondrial electron transport chain. *Science* **340**, 1567-1570 (2013).
- New ref #8. Calvo, E. et al. Functional role of respiratory supercomplexes in mice: SCAF1 relevance and segmentation of the Q_{pool}. *Sci. Adv.* **6**, eaba7509 (2020).
- New ref #9. Althoff, T., Mills, D.J., Popot, J.L. & Kühlbrandt, W. Arrangement of electron transport chain components in bovine mitochondrial supercomplex I₁III₂IV₁. *EMBO J.* **30**, 4652-4664 (2011).
- New ref #10. Szibor, M. et al. Bioenergetic consequences from xenotopic expression of a tunicate AOX in mouse mitochondria: Switch from RET and ROS to FET. *Biochim. Biophys. Acta Bioenerg.* **1861**, 148137 (2020).
- New ref #11. Trouillard, M., Meunier, B. & Rappaport, F. Questioning the functional relevance of mitochondrial supercomplexes by time-resolved analysis of the respiratory chain. *Proc. Natl. Acad. Sci. U S A.* **108**, E1027-E1034 (2011).
- New ref #12. Blaza, J.N., Serreli, R., Jones, A.J., Mohammed, K. & Hirst, J. Kinetic evidence against partitioning of the ubiquinone pool and the catalytic relevance of respiratory-chain supercomplexes. *Proc. Natl. Acad. Sci. U S A.* **111**, 15735-15740 (2014).
- New ref #13. Letts, J.A., Fiedorczuk, K. & Sazanov, L.A. The architecture of respiratory supercomplexes. *Nature* **537**, 644-648 (2016).
- New ref #14. Fedor, J.G. & Hirst, J. Mitochondrial supercomplexes do not enhance catalysis by quinone channeling. *Cell Metab.* **28**, 525-531.e4 (2018).
- New ref #15. Maranzana, E., Barbero, G., Falasca, A.I., Lenaz, G. & Genova, M.L. Mitochondrial respiratory supercomplex association limits production of reactive oxygen species from complex I. *Antioxid. Redox Signal.* **19**, 1469-1480 (2013).
- New ref #16. Lopez-Fabuel, I. et al. Complex I assembly into supercomplexes determines differential mitochondrial ROS production in neurons and astrocytes. *Proc. Natl. Acad. Sci. U S A.* **113**, 13063-13068 (2016).
- New ref #17. Guarás, A. et al. The CoQH₂/CoQ ratio serves as a sensor of respiratory chain efficiency. *Cell Rep.* **15**, 197-209 (2016).
- New ref #18. Garaude, J. et al. Mitochondrial respiratory-chain adaptations in macrophages contribute to antibacterial host defense. *Nat. Immunol.* **17**, 1037-1045 (2016).
- New ref #19. Greggio, C. et al. Enhanced respiratory chain supercomplex formation in response to exercise in human skeletal muscle. *Cell Metab.* **25**, 301-311 (2017).
- New ref #20. Balsa, E. et al. ER and nutrient stress promote assembly of respiratory chain supercomplexes through the PERK-eIF2 α axis. *Mol. Cell* **74**, 877-890.e6 (2019).

2. Line 85. “However, the molecular pathways regulating MRC supercomplex formation are largely unknown, partly owing to time-consuming and complex assays involved.” I

do not think that time-consuming assays are delaying progress in the field. Perhaps this needs to be rephased.

Ans.: We agree with the comment raised by the reviewer and have changed the phrases in lines 84-87 of the original manuscript as follows.

Old version: “However, the molecular pathways regulating MRC supercomplex formation are largely unknown, partly owing to time-consuming and complex assays involved. MRC supercomplexes are usually assayed using blue native polyacrylamide gel electrophoresis (BN-PAGE) or cryo-electron microscopy.” (lines 84-87 on page 4 of the original manuscript)

Revised version: “However, the molecular pathways regulating MRC supercomplex formation are largely unknown. While blue native polyacrylamide gel electrophoresis (BN-PAGE) or cryo-electron microscopy are usually used for analyzing the status of MRC supercomplexes, it will be further beneficial to develop an alternative rapid and simple quantitative assay for the evaluation of MRC supercomplex status, which can be applied to a medium-throughput or high-throughput screening.” (lines 99-104 on page 5 of the revised manuscript)

Results

1. Fragment about β -Lapachone starting in line 161. This compound seems to enhance mitochondrial biogenesis and therefore is not surprising that complexes and supercomplexes are also enhanced. A different issue is whether it directly promotes supercomplex assembly as stated in the text.

Ans.: As the reviewer pointed out, we added the description that β -lapachone seems to enhance mitochondrial biogenesis and increase the amounts of complexes.

“In the present study, we showed that β -lapachone can promote MRC supercomplex assembly in C2C12 myoblastic cells by our FRET-based assay. Given that β -Lapachone is an activator for mitochondrial biogenesis as reported in the literature, the enhancement of supercomplex formation may result from the increase in complexes.” (lines 339-342 on page 14 of the revised manuscript)

2. In general, compounds that enhance mitochondrial biogenesis, enhance the levels of mitochondrial complexes and supercomplexes, and have been also found to improve muscle physiology. The effect is similar to the previously-reported overexpression of PGC1-alpha. This should be mentioned in the text.

Ans.: As pointed out by the reviewer, we now mention the role of PGC-1 α on mitochondrial biogenesis, MRC supercomplex formation, and improved muscle physiology as one of the key mechanisms for MRC supercomplex formation in the revised manuscript.

“In the present study, we showed that β -lapachone can promote MRC supercomplex assembly in C2C12 myoblastic cells by our FRET-based assay. Given that β -Lapachone is an activator for mitochondrial biogenesis as reported in the literature, the enhancement of supercomplex formation may result from the increase in complexes. For example, β -lapachone has been reported to activate peroxisome proliferator-activated receptor gamma coactivator (PGC)-1 α , which is a transcriptional coactivator important for mitochondrial biogenesis and type I muscle fiber physiology⁵⁶⁻⁵⁹, and increases the levels of both MRC complex proteins and supercomplexes⁶⁰.” (lines 339-345 on page 14 of the revised manuscript)

New ref #56. Lee, M., Ban, J.J., Chung, J.Y., Im, W. & Kim, M. Amelioration of Huntington's disease phenotypes by Beta-Lapachone is associated with increases in Sirt1 expression, CREB phosphorylation and PGC-1 α deacetylation. *PLoS One* **13**, e0195968 (2018).

New ref #57. Lin, J., Handschin, C. & Spiegelman, B.M. Metabolic control through the PGC-1 family of transcription coactivators. *Cell Metab.* **1**, 361-370 (2005).

New ref #58. Kelly, D.P. & Scarpulla, R.C. Transcriptional regulatory circuits controlling mitochondrial biogenesis and function. *Genes Dev.* **18**, 357-368 (2004).

New ref #59. Lin, J. et al. Transcriptional co-activator PGC-1 alpha drives the formation of slow-twitch muscle fibres. *Nature* **418**, 797-801 (2002).

New ref #60. Srivastava, S., Diaz, F., Iommarini, L., Aure, K., Lombes, A. & Moraes, C.T. PGC-1 α /beta induced expression partially compensates for respiratory chain defects in cells from patients with mitochondrial disorders. *Hum. Mol. Genet.* **18**, 1805-1812 (2009).

3. The data regarding BAY61 and GSK143 is difficult to interpret because figure 4e shows the steady-state levels of respiratory chain proteins “using mitochondrial proteins from C2C12 myoblastic”. The WB should be done in whole cell extracts to allow for the evaluation of potential increases in mitochondrial mass.

Ans.: As suggested by the reviewer, we repeated the WB experiments using whole cell lysates of C2C12 cells treated with MNS, BAY61-3606, GSK143, and siSyk (#1 and #2). These treatments did not markedly affect the quantities of MRC complex proteins in whole cell lysates. The representative data sets are shown in Figs. 3f, 4f, and 5b of the revised manuscript. In addition, we have described these results in the revised manuscript as follows:

“We found that protein levels of NDUFB8 (complex I), RISP (complex III), and COX8A (complex IV) in the mitochondrial fraction and whole cell lysates were not markedly changed in response to MNS (Fig. 3e, f), suggesting that increased abundance of mitochondria, or mitochondrial biogenesis, may not be the primary reason for increased MRC supercomplex formation.” (lines 229-233 on page 10 of the revised manuscript)

“These additional SYK inhibitors also did not obviously increase the protein abundance of respiratory chain subunits in the mitochondrial fraction and whole cell lysates (Fig. 4e, f).” (lines 246-247 on page 11 of the revised manuscript)

“The protein concentrations of respiratory chain subunits in whole cell lysates were not markedly changed by silencing SYK expression (Fig. 5b).” (lines 258-259 on page 11 of the revised manuscript)

4. The quality of some WB is not optimal, particularly regarding the loading controls, which surely made it difficult to perform accurate quantifications. Examples are in Fig 4d and Fig 5d.

Ans.: As suggested by the reviewer, we improved the quality of BN-PAGE technique and repeated the experiments shown in Figs. 3d, 4d, and 5d of the original manuscript three times to ensure reproducibility. The results of these experiments indicate that MRC supercomplex formation is promoted by SYK inhibition, consistent with the original experiments. The representative data sets of these new experiments are now shown in Figs. 3d, 4d, and 5d of the revised manuscript.

Reviewer #3 (Remarks to the Author):

This study validates a FRET-based approach of detecting mitochondrial respiratory supercomplex formation in cells. Using this approach, drugs were screened and identified for their ability to modulate supercomplex formation using FRET as a read-out in addition to verification by BN-PAGE. These experiments led to an in vivo treatment approach that demonstrated positive effects of the drugs on locomotor functional tests in mice relative to DMSO injected controls that were associated with BN-PAGE assessed changes in supercomplex formation.

Ans.: We appreciated the valuable comments from this reviewer.

The study is interesting and logical in it's progression. Appropriate methodologies have been used overall. There are several uncertainties outlined below. The major areas of uncertainty are

Q1. whether the DMSO dose injected in vivo was too high (10%) which would require a non-DMSO control group,

Q2. whether the screening approach based on transfection approaches for FRET are considerably more advantageous than BN-PAGE (more specific elaboration of the benefit of this approach in cells is important), and

Q3. whether the assumed components of the supercomplex in question were captured in the assay as noted below.

Ans.: We appreciated the valuable comments raised by the reviewer. Regarding Q1-Q3, we addressed each of them as follows.

As for Q1, DMSO was injected to mice at a dose of 1.5 mL/kg as a vehicle in 100 μ L physiological saline. To determine the effect of DMSO on mice, a wire-hanging test was performed on mice injected with physiological saline and DMSO (twice a week for 3 weeks). The results showed no significant difference in hanging time between the DMSO-treated group and physiological saline-treated group. The results were newly added to the revised manuscript as Extended Data Fig. 5c and we described this result in the text as follows.

“There was no significant difference in hanging time between mice treated with DMSO alone and physiological saline (Extended Data Fig. 5c).” (lines 286-287 on page 12 of the revised manuscript)

Regarding Q2, the advantage of our FRET-based assay is that it can be applied to medium-throughput screens and is useful in the search for compounds that promote MRC supercomplex assembly. We have added this information as follows.

“In the present study, we established a FRET-based platform that quantifies the interactions between mitochondrial respiratory complexes I and IV, which approximates MRC supercomplex formation in live cells. This platform can be applied to medium-throughput screens as we successfully performed a screen of 1,280 pharmacologically active compounds and identified SYK inhibitors as enhancers of MRC supercomplex formation.” (lines 301-305 on page 13 of the revised manuscript)

For Q3, our FRET-based assay can detect complexes I and IV interaction in live cells and serves as an assay for physical interactions of complexes I/III₂/IV, as I/III₂/IV, I/III₂, and III₂/IV complexes are most common MRC supercomplexes in mammalian cells. However, the possibility remains that this assay may detect other types of supercomplexes as this reviewer mentioned a recent preprint that shows the existence of a megacomplex including complexes I, II, III and IV as below. We modified the description in the Results section in the revised text as shown in the answer for Query #A.

The responses to the specific comments are described below.

Specific comments:

Q #A

*Line 111-112: A very recent preprint identifies Complexes I, II, III and IV can exist as a megacomplex. While this is not peer reviewed, it introduces uncertainty regarding the assumption of the present paper that I and IV do not bind. Given this assumption guided the interpretation that FRET-based detection of I-IV must mean the supercomplex I-III₂-IV were detected, it seems that a need for clarification of the assumption would be beneficial. What confidence can be provided that I-IV detections do not reflect alternative super/mega complexes as identified in this preprint?
<https://www.biorxiv.org/content/10.1101/2022.06.26.497646v1>
Or see this link: doi: <https://doi.org/10.1101/2022.06.26.497646>*

Ans. #A: We thank the reviewer for pointing out a new study in ciliate protist *Tetrahymena thermophila* regarding megacomplex. We have changed the sentence in lines 110-112 of the original manuscript as follows:

Old version: “MRC supercomplexes consist of I/III₂/IV, I/III₂, and III₂/IV but not with I/IV. Therefore, live detection of complexes I and IV interaction serves as an assay for physical interactions of complexes I/III₂/IV in live cells.” (lines 110-112 on page 5 of the original manuscript)

Revised version: “Previous literature has defined that the complexes I/III₂/IV, I/III₂, and III₂/IV are predominant MRC supercomplexes in mammalian cells. While a possibility remains for the existence of other types of supercomplex such as I₂/III₂/IV₂ in humans and I/II/III₂/IV₂ in ciliates^{40,41}, we assume that live monitoring of complexes I and IV interaction mostly reflect the status of physical interactions of complexes I/III₂/IV in living mammalian cells.” (lines 128-132 on pages 6 of the revised manuscript)

New ref #40. Guo, R., Zong, S., Wu, M., Gu, J. & Yang, M. Architecture of human mitochondrial respiratory megacomplex I₂III₂IV₂. *Cell* **170**, 1247-1257.e12 (2017).

New ref #41. Mühleip, A. et al. Structural basis of mitochondrial membrane bending by I-II-III₂-IV₂ supercomplex. *bioRxiv* doi: <https://doi.org/10.1101/2022.06.26.497646>

Q #B

For experiments in Figure 3, was the concentration of DMSO verified to have no effect on the FRET signal? A similar question pertains to the in vivo injections in Figure 6. The methods indicate that mice were injected with 10% DMSO which seems high given lower concentrations are often used in drug studies. Comparison of DMSO treated samples and mice to saline-injections (to control for the stress of injections vs DMSO) would be important at this seemingly high DMSO dose.

Ans. #B: We thank the reviewer for valuable comments.

As suggested by the reviewer, we tested whether DMSO treatment affect the value of cFRET/donor ratio, and showed that there is no significant difference in cFRET/donor ratio between DMSO-treated cells and untreated cells. We described this results in the text as follows.

“The value of cFRET/donor significantly increased by treatment of β -lapachone at a concentration of 1 μ M, while vehicle (DMSO) treatment had no significant effect (Extended Data Fig. 2b, c).” (lines 194-196 on page 9 of the revised manuscript)

DMSO was injected at a dose of 1.5 mL/kg as a vehicle in 100 μ L physiological saline. To examine the effect of DMSO dosage in mice, we performed wire hanging test using mice injected with this concentration of DMSO or physiological saline twice a week for 3 weeks. As a result, we showed that there was no

significant difference in hanging time between DMSO- and saline-treated groups. We added this result to the revised manuscript as follows.

“There was no significant difference in hanging time between mice treated with DMSO alone and physiological saline (Extended Data Fig. 5c).” (lines 286-287 on page 12 of the revised manuscript)

Q #C

Figure 3/Supplemental Table S1: what was the procedure for deciding on the concentration to use for each candidate compound for the initial screen (40uM)? The finding that most of the 222 chemicals with a FRET value above 0.6 caused cell death or change in cell morphology might not exclude their potential role at lower concentrations, for example. Could the authors comment on the possibility that some of these compounds might still have potential efficacy at doses not tested in this study?

Ans. #C: We appreciated the valuable comments raised by the reviewer. For cell-based screening, we have previously used a different bioactive library at 20 μ M in C2C12 myotubes [Nat Biotechnol 26:343-351, 2008] and LOPAC library at 40 μ M in macrophages (ref #50). We added this information in the revised Results section as below (underline highlighted).

“In the primary screen, the FRET signals in cells treated with these chemicals (40 μ M each), β -lapachone (40 μ M), and vehicle (DMSO) were evaluated using an imaging cytometer. The concentration of compounds was determined based on our previous experience using a LOPAC 1280 library⁵⁰.” (lines 205-208 on page 9 of the revised manuscript)

New ref #50. Tran, U. T. & Kitami, T. Niclosamide activates the NLRP3 inflammasome by intracellular acidification and mitochondrial inhibition. *Commun. Biol.* **2**, 2 (2019).

We agree that in hindsight, 40 μ M may have been at the higher end of the typical range (10 to 40 μ M) used in cell-based screening. However, a compound with desirable effect in cell culture at 10 μ M but toxic at 40 μ M might run into a risk of being toxic or difficult to work with when administered *in vivo*. Nonetheless, we agree on this point and added this point in the Discussion section in the revised text (underline highlighted) that some of the compounds may have been missed due to the higher concentration used in our screen.

Revised version: “This platform can be applied to medium-throughput screens as we successfully performed a screen of 1,280 pharmacologically active compounds and identified SYK inhibitors as enhancers of MRC supercomplex formation. While we set 40 μ M as a threshold concentration of compounds in the primary screening, a possibility

remains that we identified more compounds which promote supercomplex formation at lower concentrations.” (lines 303-307 on page 13 of the revised manuscript)

Q #D

Fig 3 figure legend might list the duration of treatment for MNS

Ans. #D: As suggested by the reviewer, we have added the duration of treatment for MNS (underline highlighted) to the legends of Fig. 3 in the revised manuscript as below.

“Fig. 3. 3,4-methylenedioxy- β -nitrostyrene (MNS) was identified as a compound inducing mitochondrial respiratory chain supercomplex formation using chemical library screen with FRET imaging. a, Scheme of the high-throughput screen procedures using imaging cytometer. Compounds which induced high cFRET/donor value intensity in C2C12 myoblastic cells stably co-expressing FRET pair of NDUFB8-AcGFP and COX8A-DsRed-Monomer were selected for further analysis. b, Chemical structures of the final validated compound from the screening assay, MNS. c, cFRET/donor ratio of C2C12 myoblastic cells stably co-expressing FRET pair of NDUFB8-AcGFP and COX8A-DsRed-Monomer treated by different concentrations of MNS for 24 h ($n = 3$). EC_{50} , half maximal effective concentration. d, BN-PAGE of mitochondrial proteins from C2C12 myoblastic cells treated with MNS (1 μ M) or vehicle (DMSO) for 24 h. Positions corresponding to mitochondrial supercomplexes I/III₂/IV_n, I/III₂/IV, I/III₂, III₂/IV, and dimerized complex III (III₂) are indicated. Western blot analysis was performed with antibody for UQCRC2 of complex III. FP70 protein was blotted as an internal control. IB, immunoblot. e, SDS-PAGE of proteins from mitochondrial fraction of C2C12 myoblastic cells treated with MNS (1 μ M) or vehicle (DMSO) for 24 h. Western blot analysis was performed with antibodies for NDUFB8 of complex I, COX8A of complex IV, and RISP of complex III. FP70 protein was blotted as an internal control. f, SDS-PAGE of mitochondrial proteins from whole cell lysates of C2C12 myoblastic cells treated with MNS (1 μ M) or vehicle (DMSO) for 24 h. Western blot analysis was performed with antibodies for NDUFB8, COX8A, RISP, FP70, and ATP5A. β -Actin was blotted as an internal control. g, Mitochondrial respiration of C2C12 myoblastic cells treated with MNS (1 μ M) or DMSO for 24 h was measured using Seahorse XFp Cell Mito Stress Test. Oxygen consumption rates (OCRs) are shown. Data are presented as means \pm SE ($n = 3$). Statistical analyses were applied to basal respiration (before oligomycin injection), ATP production (between oligomycin and FCCP injection), and maximal respiration (between FCCP and rotenone/antimycin A injection). Basal respiration was measured as the OCR before oligomycin injection minus the non-mitochondrial respiration. ATP production was measured as the OCR prior to oligomycin injection minus the OCR between oligomycin and FCCP injection. Maximal respiration was measured as the OCR after FCCP addition minus the non-mitochondrial respiration. * $P < 0.05$ (Student’s t -test).”

Q #E

Fig 3f – the ATP synthesis component of the OCR trace could be determined and tested for differences between MNS and control (reduction with oligomycin; the actual absolute loss of OCR with oligomycin rather than the new steady state which is more attributable to leak-induced respiration). This applies to the remaining OCR graphs in subsequent figures as well.

Ans. #E: We appreciated the valuable comments from the reviewer. We have now calculated the absolute change in OCR between basal respiration and oligomycin-suppressed respiration as a measure of “ATP synthesis component” for Figs. 3f, 4f, 4g, 5e, and 5f in the original manuscript. We could not detect significant increases in ATP synthesis component for GSK143 and siSyk. The discussion of this point is provided in detail below in response comments (Q #G).

Q #F

In general, the supercomplex blots (Fig 3D, 4D as examples) do not convey consistency of findings given these are n=1. How reproducible are these findings?

Ans. #F: As suggested by the reviewer, we improved our BN-PAGE technique and repeated the experiments shown in Figs. 3d, 4d, and 5d of the original manuscript three times to ensure reproducibility. The results of these new experiments indicate that MRC supercomplex formation is promoted by SYK inhibition, consistent with the original experiments. The representative data from these new experiments are shown in Figs. 3d, 4d, and 5d of the revised manuscript.

Q #G

Fig 4g – the finding that GSK143 only increased ‘maximal’ OCR can be discussed in terms of its utility as a index of maximal flux through the ETS vs the lack of physiological relevance of this measure given it is an uncoupled phenomenon and there was no change in the apparent ATP synthesis component of basal respiration (although this requires the t-test on the absolute drop in OCR with oligomycin). A similar question pertains to Fig 5F.

Ans. #G: As the reviewer pointed out, GSK143 and siSyk treatment resulted in significant increases in maximal respiration but no change in ATP synthesis-linked respiration in C2C12 cells (Figs. 4h, 5e, and 5f). The enhanced OCR in maximal respiration relative to basal respiration indicates that mitochondrial “spare respiratory capacity” (SRC) was increased. SRC is considered as an important parameter for the adaptation to stress condition, and SRC deficiency is associated with neurodegenerative diseases and cardiac diseases. Thus, SYK

inhibition may have an effect on muscle function by increasing the stress adaptative capacity of mitochondrial respiration.

In addition, we checked the level of an ATP synthase protein, ATP5A, in C2C12 cells treated with SYK inhibitors and siSyk. We found that the protein levels were not markedly changed, further confirming that ATP synthesis component of basal respiration was not obviously enhanced.

We have added this point in the revised manuscript as follows:

“The OCR measurements in C2C12 cells treated with SYK inhibitors and siSyk showed that SYK inhibitors and siSyk enhanced the maximal respiration. The higher OCR in maximal respiration results in the enhanced mitochondrial spare respiratory capacity (SRC). SRC is considered as an important parameter for the adaptation to stress condition, and SRC deficiency is speculated to be associated with neurodegenerative diseases and cardiac diseases⁶¹⁻⁶³. Thus, SYK inhibition may affect muscle function partly by promoting the stress adaptative capacity of mitochondria. Importantly, MNS and BAY61-3606 also increased both the basal and ATP synthesis component of respiration (oligomycin-suppressed OCR), suggesting that additional targets of MNS and BAY61-3606 may be responsible for this additional effect on ATP synthesis. Moreover, we have checked the level of an ATP synthase protein, ATP5A in C2C12 cells treated with SYK inhibitors and siSyk, and found that SYK inhibition did not markedly affect the expression of ATP5A. This result suggests that the difference in the effects of SYK inhibitors on mitochondrial respiration may be independent of ATP synthase biosynthesis. For better understanding of these points, it will be useful to reveal the mechanism by which SYK regulates MRC supercomplex formation.” (lines 350-364 on page 15 of the revised manuscript).

New ref #61. Marchetti, P., Fovez, Q., Germain, N., Khamari, R. & Kluza, J. Mitochondrial spare respiratory capacity: Mechanisms, regulation, and significance in non-transformed and cancer cells. *FASEB J.* 34, 13106-13124 (2020).

New ref #62. Desler, C., Hansen, T.L., Frederiksen, J.B., Marcker, M.L., Singh, K.K. & Juel Rasmussen, L. Is there a link between mitochondrial reserve respiratory capacity and aging? *J. Aging Res.* 2012, 192503 (2012).

New ref #63. Yamamoto, H. et al. Amla enhances mitochondrial spare respiratory capacity by increasing mitochondrial biogenesis and antioxidant systems in a murine skeletal muscle cell line. *Oxid. Med. Cell. Longev.* 2016, 1735841 (2016).

Q #H

Fig 6 - <60 seconds cage hang time is quite low for a healthy wildtype mouse, although perhaps this is strain dependent. Some strains can hold on for over 150 seconds which often leads to the requirement to set a maximal time for the test (it seems a similar

threshold was set in extended figure 5d for the GSK group). Similar to the question above, did DMSO affect cage hang time? Does this time not seem low?

Ans. #H: As pointed out by the reviewer, the mice used in Fig. 6 are the same strain (DBA/2CrSlc) as the mice used in Extended Data Fig. 5, but the hanging time was shorter. All of these mice were maintained under the same conditions at the animal facility of the Tokyo Metropolitan Institute of Gerontology, but were purchased at different times. As described in Ans #B, we have shown that DMSO treatment has no significant effect on the hanging time of mice, suggesting that DMSO treatment is not the cause of short hanging time. Although the mice used in Fig. 6 have short hanging times, MNS treatment increased hanging time as well as running time and distance in the treadmill test, suggesting that these mice remain susceptible to SYK inhibition, and thus the results obtained from the experiments with these mice are useful for studying the effects of SYK on exercise performance.

Q #I

Conclusion line 369 – the basis for determining effective preventative treatments for muscle weakness could be rephrased to be more speculative and broad as no model of muscle weakness was assessed.

Ans. #I: As the reviewer pointed out, we change the description in line 369 of the original manuscript as follows.

Old version: “Our study provides insights into the regulatory mechanism of MRC supercomplex formation and provides a basis for the effective prevention or treatment of conditions in which skeletal muscle weakness is involved.” (line 369 on page 15 of the original manuscript)

Revised version: “Our study provides insights into the regulatory mechanism of MRC supercomplex formation and would be beneficial in clinical application settings, such as in the development of effective prevention or treatment for conditions involving muscle weakness.” (lines 461-464 on page 19 of the revised manuscript).

Q #J

Model: the DBA/2 line of mice is known to have a natural mutation in TGF-beta binding protein 4 leading to elevated activity of TGF-beta. This model has elevated fibrosis and impaired regenerative satellite cell capacities amongst other distinct attributes. Can the authors comment on the impact of this natural mutation on their data? However, I am not sure if DBA/2 is the same strain as DBA/2CrSlc used in this study.

Ans. #J: We thank this reviewer for raising an important point in terms of mouse strain-specific background. As commented by the reviewer, DBA/2CrSlc that we used in this study is a wild-type strain of DBA/2 line with amino acids deletion in *Ltbp4* gene. This genetic background is relevant to muscular dystrophy susceptibility and fibrosis because of the mutant LTBP4 protein has reduced activity to sequester TGF- β . Because SYK inhibition was shown to decrease TGF- β expression and signaling in some tissues, we assume that the effects of SYK inhibitors on muscle activity can be modulated by LTBP4 expression or TGF- β signaling. We added this point to the Discussion section in the revised manuscript as below.

“Notably, DBA/2CrSlc used in the present study is a wild-type strain of DBA/2 line, which possesses 12 amino acids deletion in the proline-rich region of *Ltbp4* gene. This mutant LTBP4 protein exhibits reduced activity to sequester TGF- β , leading to an enhanced activity of the cytokine. This genetic background is relevant to muscular dystrophy susceptibility, thus the introduction of null allele of *dystrophin-associated protein γ -sarcoglycan* (*Sgcg*) into the DBA/2 background exhibited a more severe muscular dystrophy phenotype than the 129T strain⁸⁰. Intriguingly, SYK inhibition has been shown to decrease the expression of TGF- β in skin and lung and to repress fibrosis in these tissues⁸¹. While we used only DBA/2CrSlc strain to evaluate the effects of SYK inhibitors on muscle activity, we consider that this strain may be a suitable model for evaluating the impact of SYK inhibitors in muscles because the strain background primarily exhibits susceptibility to muscular dystrophy and fibrosis. A recent study also revealed that anti-LTBP4 treatment reduced muscle fibrosis and enhanced muscle force production⁸², thus, further study will elucidate whether the effects of SYK inhibitors on muscle activity can be modulated by LTBP4 expression or TGF- β signaling. Considering that the muscle fibrosis and impaired regenerative satellite cell capacities are often observed in aged patients, it is tempting to speculate that SYK inhibition can be more beneficial for patients with the enhanced TGF- β activity in muscles.” (lines 429-445 on page 18 of the revised manuscript)

New ref #80. Heydemann, A. et al. Latent TGF-beta-binding protein 4 modifies muscular dystrophy in mice. *J. Clin. Invest.* **119**, 3703-3712 (2009).

New ref #81. Pamuk, O.N., et al. Spleen tyrosine kinase (Syk) inhibitor fostamatinib limits tissue damage and fibrosis in a bleomycin-induced scleroderma mouse model. *Clin. Exp. Rheumatol.* **33**, S15-22 (2015).

New ref #82. Demonbreun, A.R. et al. Anti-latent TGF β binding protein 4 antibody improves muscle function and reduces muscle fibrosis in muscular dystrophy. *Sci. Transl. Med.* **13**, eabf0376 (2021).

Q #K

Functional tests: more detail on the cage hang time could be included. Were measures

made in duplicate or single tests? Was 150 seconds set as the maximum time as implied in the extended data figure 5?

Ans. #K: We appreciated the valuable comments raised by the reviewer. We measured the hanging time twice and adopted the longer time. The maximum time was set to 150 seconds. We have added these points to the revised manuscript.

“The hanging time was measured twice and the longer time was adopted for the data. The maximum measuring time was set to 150 seconds.” (lines 919-920 on page 38 of the revised manuscript)

Q #L

TEM: how many sections were assessed per sample and how many images per section were quantified? Extended fig 4E gives an ‘n’ but does not indicate if these are sections or areas within a single section (likely the latter, but how many sections?).

Ans. #L: We thank the reviewer for valuable comments. The “*n*” shown in Extended Data Figs. 4e and 5l indicates the number of mitochondria whose size was measured. In Extended Data Fig. 4e, we observed sections of soleus muscles from two mice treated with DMSO and two mice treated with MNS. For DMSO-treated mice, five sections were prepared from the first mouse and three sections from the second mouse for observation of mitochondria. For the MNS-treated mice, three sections were prepared from the first mouse and three sections were prepared from the second mouse. In Extended Data Fig. 5l, we observed sections of soleus muscles from one DMSO-treated mouse and two mice treated with BAY61-3606 and two mice treated with GSK143. Three sections were prepared from one DMSO-treated mouse and two sections from each of the two BAY61-3606 mice and two sections from each of the two GSK143-treated mice. We have added this information to the legends of Extended Data Fig. 4e and 5l as follows.

“Sections of soleus muscles from two mice treated with DMSO and two mice treated with MNS were used for the quantification of mitochondrial size. For DMSO-treated mice, five sections were prepared from the first mouse and three sections from the second mouse. For the MNS-treated mice, three sections were prepared from each of two mice.” (lines 1138-1142 on page 59 on the revised manuscript)

“Sections of soleus muscles from one DMSO-treated mouse, two BAY61-3606-treated mice, and two GSK143-treated mice were used for quantification of mitochondrial size. Three sections were prepared from one DMSO-treated mouse, and two sections per

mouse were prepared from two BAY61-3606-treated and two GSK143-treated mice.” (lines 1171-1175 on page 61 of the revised manuscript)

Q #M

Was normality verified before using parametric tests where appropriate?

Ans. #M: In terms of normality in statistical analysis, we verified it by Shapiro-Wilk test. We added the information in the Online Methods section of the revised text as follows.

“The normality of the experimental data was examined by Shapiro-Wilk test.” (lines 951-952 on page 39 of the revised manuscript)

We analyzed the data in Figs. 6a and 6b with Mann-Whitney *U*-test, the data in Extended Data Figs. 4e and 5l with median test, and the data in Extended Data Figs. 5b with Kruskal-Wallis test and post-hoc Mann-Whitney *U*-test. This statistical analysis did not affect the presence or absence of significant differences. We modified the figure of Extended Data Fig. 5b to reflect the change in *P*-values (DMSO vs GSK, *P* = 0.0006). We added the information regarding this analysis to the legends of each figure as follows.

“Fig. 6. Increased exercise performance in mice treated with MNS.

a, Body weights of mice after intraperitoneal injection of MNS (4 mg/kg) or DMSO twice a week for 5 weeks. Data are presented as means ± SE (*n* = 8). n.s., not significant (Mann-Whitney *U*-test). **b**, Results of wire hanging test after injection of MNS or DMSO for 3 weeks. Data are presented as means ± SE (*n* = 8). ***P* < 0.01 (Mann-Whitney *U*-test).”

“Extended Data Fig. 4. MNS treatment did not affect muscle weight.

e, The size of the mitochondria in the soleus muscle from MNS- or DMSO-treated mice were quantified by transmission electron microscopic examination. Indicated number of mitochondria was evaluated. Sections of soleus muscles from two mice treated with DMSO and two mice treated with MNS were used for quantification of mitochondrial size. For DMSO-treated mice, five sections were prepared from the first mouse and three sections from the second mouse. For the MNS-treated mice, three sections were prepared from each of two mice. Data are presented as means ± SE. n.s., not significant (median test)”

“Extended Data Fig. 5. Increased exercise performance in mice treated with SYK inhibitors.

b, Results of wire hanging test after injection of DMSO, BAY61-3606, or GSK143 for 3 weeks. Data are presented as means \pm SE. * $P < 0.05$, *** $P < 0.001$ (Kruskal-Wallis test and post-hoc Mann-Whitney U -test).

l, The size of the mitochondria in the soleus muscle from DMSO-, BAY61-3606- or GSK143-treated mice were quantified by transmission electron microscopic examination. Indicated number of mitochondria was evaluated. Sections of soleus muscles from one DMSO-treated mouse, two BAY61-3606-treated mice, and two GSK143-treated mice were used for quantification of mitochondrial size. Three sections were prepared from one DMSO-treated mouse, and two sections per mouse were prepared from two BAY61-3606-treated and two GSK143-treated mice. Data are presented as means \pm SE. n.s., not significant (median test).”

Q #N

OCR: Was the drug present during the OCR assay in the cell experiments, or were the cells pre-treated with the drug without the drug present in the media during the actual assay?

Ans. #N: We appreciated the comments provided by the reviewer. After treating the cells with SYK inhibitors and siSyk, the medium was replaced with XF Base Medium Minimal DMEM (containing substrates) free of these reagents and then the OCRs were measured with an XF Extracellular Flux Analyzer. We have added this information (underline highlighted) to the revised manuscript as below.

“One hour before the assay, the culture medium was replaced with XF Base Medium Minimal DMEM containing the substrates (1 mM sodium pyruvate, 2 mM glutamine, and 25 mM glucose) (Seahorse Biosciences), in which SYK inhibitors and siSyk were not added.” (lines 898-901 on page 37 of the revised manuscript)

REVIEWERS' COMMENTS

Reviewer #2 (Remarks to the Author):

The authors have now improved the WB data, modified the text where needed, and responded to all my queries. I believe the manuscript is now suitable for publication.

Reviewer #3 (Remarks to the Author):

The authors have included new data that shows no effect of their DMSO dose on hang time and FRET signal. 10% is still risky given accumulating evidence shows toxic effects in vivo beyond 2%. It is unfortunate that each measure was not verified with saline vs 10% DMSO prior to beginning such a large study, but the new data somewhat increases confidence that the vehicle may not be affecting the other measures (running time, muscle weights, etc).

See:

DOI: 10.1096/fj.13-235440

<https://doi.org/10.1038/s41598-022-07706-2>

This is a comment for the authors to consider for future directions given the growing concern in the literature. The two controls in the new data nonetheless provide some confidence in potential lack of effect of DMSO on their measures.

Second, the authors indicate that they retrospectively analyzed the oligomycin-suppression of respiration (ATP synthesis component) and did not find an effect of the drugs. They added a new section to the discussion that presents a more specific and reasonable interpretation of the effect on maximal respiration in the presence of an uncoupler: the drugs increase spare respiratory capacity as a measure of the adaptive potential for mitochondria to increase the capacity for ATP synthesis in the future under stress, if needed, and that this fits the observation of altered supercomplex formation. This is a useful addition to the discussion. The acknowledgement that there was no effect of the drug on the ATP synthesis component (oligomycin step of the protocol) is not matched to new analyses or statistics in the figures nor is this stated in the results text. So, the discussion and interpretation is useful, but the analyses is not shown. The trace is shown, but the new results text and more clear referral to the figures for drugs showing a lack of effect on the ATP synthesis component would improve clarity for the non-expert reader.

Third, the authors rebutted with the following statement regarding BN-PAGE:

'As suggested by the reviewer, we improved our BN-PAGE technique and repeated the experiments shown in Figs. 3d, 4d, and 5d of the original manuscript three times to ensure reproducibility. The results of these new experiments indicate that MRC supercomplex formation is promoted by SYK

inhibition, consistent with the original experiments. The representative data from these new experiments are shown in Figs. 3d, 4d, and 5d of the revised manuscript. '

Where is this data? Where is there statement in the manuscript that the findings were reproducible?

Responses to Reviewers

Reviewer #2 (Remarks to the Author):

The authors have now improved the WB data, modified the text where needed, and responded to all my queries. I believe the manuscript is now suitable for publication.

Ans.: We sincerely appreciate valuable comments raised by this reviewer. We are happy that our manuscript is now suitable for publication.

Reviewer #3 (Remarks to the Author):

Q1. The authors have included new data that shows no effect of their DMSO dose on hang time and FRET signal. 10% is still risky given accumulating evidence shows toxic effects in vivo beyond 2%. It is unfortunate that each measure was not verified with saline vs 10% DMSO prior to beginning such a large study, but the new data somewhat increases confidence that the vehicle may not be affecting the other measures (running time, muscle weights, etc).

See:

DOI: 10.1096/fj.13-235440

<https://doi.org/10.1038/s41598-022-07706-2>

This is a comment for the authors to consider for future directions given the growing concern in the literature. The two controls in the new data nonetheless provide some confidence in potential lack of effect of DMSO on their measures.

Ans.: We appreciate this outstanding reviewer for her/his valuable comments. As suggested by the reviewer, we will carefully determine an optimal concentration of DMSO before starting a large study in the future.

Q2. Second, the authors indicate that they retrospectively analyzed the oligomycin-suppression of respiration (ATP synthesis component) and did not find an effect of the drugs. They added a new section to the discussion that presents a more specific and reasonable interpretation of the effect on maximal respiration in the presence of an uncoupler: the drugs increase spare respiratory capacity as a measure of the adaptive

potential for mitochondria to increase the capacity for ATP synthesis in the future under stress, if needed, and that this fits the observation of altered supercomplex formation. This is a useful addition to the discussion. The acknowledgement that there was no effect of the drug on the ATP synthesis component (oligomycin step of the protocol) is not matched to new analyses or statistics in the figures nor is this stated in the results text. So, the discussion and interpretation is useful, but the analyses is not shown. The trace is shown, but the new results text and more clear referral to the figures for drugs showing a lack of effect on the ATP synthesis component would improve clarity for the non-expert reader.

Ans.:

As suggested by this reviewer, we added the following to the text in the revised Results section as below, showing that GSK143 (Fig. 4h) and siSYK (Fig. 5e and 5f) did not significantly affect the ATP synthesis component of respiration.

“In addition, GSK143 did not significantly affect the ATP synthesis component of respiration (Fig. 4h).” (lines 251-252 on page 11)

“Furthermore, siSyk #1 and #2 treatment resulted in enhanced maximal oxygen consumption rates, while did not significantly alter the ATP synthesis component of respiration and ECARs (Fig. 5e, f, Supplementary Fig. 4d, e).” (lines 263-265 on page 11)

Q3. Third, the authors rebutted with the following statement regarding BN-PAGE:

'As suggested by the reviewer, we improved our BN-PAGE technique and repeated the experiments shown in Figs. 3d, 4d, and 5d of the original manuscript three times to ensure reproducibility. The results of these new experiments indicate that MRC supercomplex formation is promoted by SYK inhibition, consistent with the original experiments. The representative data from these new experiments are shown in Figs. 3d, 4d, and 5d of the revised manuscript. '

Where is this data? Where is there statement in the manuscript that the findings were reproducible?

Ans.:

In terms of replicated data corresponding to the representative results shown in Figs. 3d, 4d, and 5d, we showed two more series of independent experimental data as new Supplementary Fig. 3a-e in the revised manuscript. We added the description that each BN-PAGE analysis repeated three times showed the enhancement of supercomplex assembly in response to SYK inhibitors as follows:

“We evaluated MRC supercomplex formation based on BN-PAGE for three times and identified MNS as a chemical that promotes supercomplex formation (Fig. 3d, Supplementary Fig. 3a, b).” (lines 228-230 on page 10)

“We performed BN-PAGE for three times and showed that BAY61-3606 and GSK143 enhance MRC supercomplex formation (Fig. 4d, Supplementary Fig. 3a, c).” (lines 245-247 on pages 10-11)

“We performed BN-PAGE for three times and showed that siSyk #1 and #2 treatment enhanced MRC supercomplex formation in C2C12 myoblastic cells (Fig. 5d, Supplementary Fig. 3d, e).” (lines 259-261 on page 11)